# HMGA2 as a prognostic and immune biomarker in hepatocellular carcinoma: Comprehensive analysis of the HMG family and experiments validation

**Qiangqiang Zhong**[1,2☯]**, Baokang Zhao**[1,3☯]**, Xiao She**[4]**, Xiangjie Liu**[3,5]*

**1** Department of Gastroenterology, Liyuan Hospital, Tongji Medical College, Huazhong University of Science and Technology, Wuhan, PR China, **2** Laboratory of Metabolic Abnormalities and Vascular Aging Huazhong University of Science and Technology, Wuhan, PR China, **3** Department of Geriatrics, Liyuan Hospital, Tongji Medical College, Huazhong University of Science and Technology, Wuhan, PR China, **4** Department of Gastroenterology, Xi'an Jiaotong University Second Affiliated Hospital, Xi'an, PR China, **5** Key Laboratory of Vascular Aging, Ministry of Education, Tongji Hospital, Tongji Medical College, Huazhong University of Science and Technology, Wuhan, PR China

☯ These authors contributed equally to this work.
* liuxiangjie1968@126.com

**Data Availability Statement:** The datasets generated from The TCGA database (https://portal.gdc.cancer.gov) and GTEx database (https://

## Abstract

The molecular mechanisms underlying hepatocellular carcinoma (HCC) are complex and not fully understood. This study aims to explore the expression and clinical significance of High Mobility Group (HMG) proteins in HCC to identify potential prognostic biomarkers and therapeutic targets. Bioinformatic analyses were performed using data from The Cancer Genome Atlas (TCGA) and other databases. Expression levels of HMGs were validated in HCC cell lines using qRT-PCR, and functional studies were conducted by knocking down HMGA2.HMG family members, particularly HMGA1, HMGA2, HMGB2, and HMGN1, were significantly upregulated in HCC tissues compared to normal tissues. High expression levels of these proteins were associated with poor overall survival and disease-specific survival in HCC patients. Knockdown of HMGA2 in HCC cell lines led to reduced cell proliferation, migration, and invasion. HMGA2, along with other HMG family members, emerges as a potential prognostic biomarker and therapeutic target in HCC. This study provides new insights into the role of HMG proteins in HCC progression.

## Introduction

Liver cancer stands out as one of the prevailing malignancies, witnessing around 90,600 fresh cases and nearly 830,000 associated fatalities in 2020. It holds the sixth position among the most frequent cancer types and claims the third spot for global cancer-linked mortalities [1]. HCC is a complex malignancy influenced by both environmental and dietary factors, and the precise molecular pathogenesis remains incompletely understood. While advancements in liver cancer treatment have led to a gradual enhancement in patient prognosis, the overall

genome.ucsc.edu/) during and/or analyzed during the current study are publicly available. All raw datas including codes can be obtained from the following URL:https://pan.quark.cn/s/cea0185cb9c3.

**Funding:** This research has been facilitated by the generous grant from the National Key Research and Development Program of China (2020YFC2008904). The funder had no role in study design, data collection and analysis, decision to publish, or preparation of the manuscript.

**Competing interests:** Our authors of this paper confirmed that they have no conflicts of interest pertaining to the conducted study.

survival of individuals with liver cancer continues to present challenges [2]. Over the past decade, various molecular targets have been identified for the treatment of HCC, including VEGF (vascular endothelial growth factor), PD-1/PD-L1 (programmed death-ligand 1), and EGFR (epidermal growth factor receptor) [3]. These targets have been the basis for the development of targeted therapies such as sorafenib, regorafenib, and immune checkpoint inhibitors [4, 5]. However, despite these advances, the efficacy of these treatments is often limited due to drug resistance, heterogeneous tumor biology, and the complexity of HCC pathogenesis [6]. For instance, resistance to sorafenib has been observed in many patients, and the overall survival benefit provided by current therapies remains modest. Additionally, the genetic and molecular diversity within HCC tumors poses significant challenges in identifying universally effective treatment targets [7]. Given these limitations, there is an urgent need to explore novel biomarkers and therapeutic targets that can overcome these drawbacks and provide more effective treatment options for HCC patients. High Mobility Group (HMG) proteins, known for their roles in chromatin remodeling and gene regulation, have recently emerged as potential players in cancer progression [8]. However, their specific involvement in HCC has not been comprehensively evaluated. This study aims to fill this gap by investigating the expression and clinical significance of HMG family members in HCC, with the goal of identifying new prognostic biomarkers and therapeutic targets.

In recent times, molecular targeted therapy has emerged as a clinical approach for addressing diverse cancers, encompassing liver cancer, by providing heightened selectivity and specificity in treatment. Thus, the development of more targeted drugs aiming at specific pathways holds significant importance for the clinical treatment of hepatocellular carcinoma [3, 9]. Only through in-depth research into the molecular mechanisms of liver cancer pathogenesis can more effective targets be identified to develop novel targeted therapies.

High Mobility Group proteins (HMGs) were initially isolated and characterized in 1973 from the thymus of calves [10]. HMGs could be categorized into three distinct subfamilies, which named HMGA, HMGB, and HMGN [11]. Evident across a broad spectrum of eukaryotic cells, high-mobility group proteins (HMGs) function as chromosomal proteins, with content ranking second only to histones. Their primary role revolves around gene transcription regulation and the alteration of chromatin fiber structure and activity [12, 13]. The configuration of chromatin fibers assumes a pivotal role in governing gene expression precision, with histone and DNA chemical modifications serving as crucial agents in epigenetic regulation. Consequently, these proteins are deemed influential participants in the process of tumorigenesis.

The High Mobility Group A protein (HMGA) exhibits robust expression during embryonic tissue development, while being either absent or minimally expressed in adult tissue cells. However, its presence becomes heightened in an array of both malignant and benign tumors [14]. The overexpression of HMGA stands as a hallmark feature in numerous malignant and benign tumors, encompassing the majority of malignant tumors and lipomas [15, 16]. The HMGA subfamily includes HMGA1 and HMGA2. HMGA1, as a highly conserved non-histone protein, can interact with multiple transcription factors to control the activation and transcription of cancer-related genes [17]. HMGA1 is a chromatin-binding protein that plays a crucial role in embryonic development, differentiation, and tumorigenesis. Existing literature reports indicate that inhibiting HMGA1 expression can disrupt the cell cycle at the S phase, delay cell entry into the G2/M phase, and induce cell apoptosis. It can impact tumor metastasis through various mechanisms, including EMT [18]. Numerous studies have demonstrated that HMGA1 facilitates tumorigenesis by impeding the expression of the transformation-related protein 53, commonly referred to as p53, along with disrupting the expression of the cell cycle protein RB1 [19]. Likewise, contemporary investigations have illuminated HMGA2's

analogous involvement in tumor development. HMGA2 assumes the capacity to mediate transforming growth factor-beta (TGF-β) signaling, thereby exerting regulatory influence over crucial processes encompassing epithelial differentiation, tumor invasion, and metastasis [20]. Furthermore, HMGA2 can function as a target gene of various microRNAs in the development of ESCC. For instance, Hsa-circ-10006948 drives the advancement of esophageal squamous cell carcinoma (ESCC) and triggers epithelial-mesenchymal transition by orchestrating the miR-490-3p/HMGA2 axis. Likewise, the circRNA Circ-CCND1 exacerbates hepatocellular carcinoma by modulating the miR-497-5p/HMGA2 axis [21]. HMGA2 experiences dysregulation within diverse malignant tumor tissues and is duly recognized as an oncogene intricately linked to the malignancy and prognostication of neoplastic conditions.

HMGB proteins are a common nuclear protein in eukaryotic cells, secreted by various stem cells, and participate in kinds of pathophysiological activities, including inflammation, thrombosis, tissue regeneration, etc [22, 23]. The HMGB subfamily comprises HMGB1, HMGB2, and HMGB3. HMGBl belongs to the HMGB protein family, which is abundant and highly conserved in this family and exists in most eukaryotic cells. As a DNA partner, HMGB1 engages in cellular nucleus processes encompassing DNA replication, transcription, and chromatin remodeling, regulating DNA damage repair and maintaining genome stability. Extracellular HMGBl has the capacity to induce the recruitment of inflammatory cells, activate macrophages and endothelial cells, and trigger the release of cytokines, causing an inflammatory response, stimulating tumor cell proliferation, angiogenesis, endothelial cell transformation, invasion, and metastasis [24]. HMGBl not only acts as an inflammatory factor involved in tumor occurrence but also exacerbates inflammation-related immune suppression produced by tumor cells, further promoting tumor progression. Hence, the inhibition of HMGB1 synthesis and release, or the blockade of the signal pathway involving HMGB1 binding to its receptor, holds the potential to mitigate the incidence of associated disorders [25]. Both HMGB2 and HMGB1 are relatively conserved members of the HMGB family. Compared with HMGB1, the research on HMGB2 is still not very thorough. Scholars have ascertained that HMGB2 exhibits heightened expression during embryonic development, while predominantly manifesting in adult testes and lymphoid organs [26]. Recently, some researchers have also revealed that HMGB2 gets involved in tumor occurrence and progression, with high expression in various malignant tumors, and also affects the prognosis of tumor patients to some extent. Given that HMGB2 has overlapping functional domains with HMGB1 and plays a certain role in the malignant progression of various tumors, this study speculates that HMGB2 also has a certain impact on the malignancy of liver cancer. HMGB3 is mainly associated with DNA repair, cell transcription, and cancer development [27]. In addition, HMGB3 regulates cell proliferation, migration, invasion, immune escape, and apoptosis by affecting intercellular signaling [28–30].

The non-histone chromatin architectural protein family known as high mobility group nucleosome-binding proteins (HMGNs) finds extensive expression within the cell nucleus of vertebrate cells [31]. The HMGN gene family encompasses five distinct members, denoted as HMGN1, HMGN2, HMGN3, HMGN4, and HMGN5, which can specifically bind to nucleosomes. Functioning as chromatin-binding proteins within cellular contexts, every constituent of the HMGN family partakes in a spectrum of cellular functions, including DNA replication, gene expression regulation, and organ development. HMGN1 and HMGN2 significantly contribute to the modulation of organ development and maturation by orchestrating the regulation of specific genes or proteins. Additionally, they are involved in eliciting tumor immune responses [32, 33]. Human HMGN3 is most abundantly expressed in the pancreas and pituitary, and studies have shown that HMGN3 is associated with resistance to anticancer drugs such as camptothecin and paclitaxel in human liver cancer-derived cell lines, and this drug

resistance is related to the expression of transcription factors such as NFκB and AP1 [34], although the specific mechanisms remain unclear. Currently, research on HMGN4 is relatively limited, and it is expected that more studies will explore its relevance to diseases, thereby identifying additional clinical disease treatment strategies. Overexpression of HMGN5 is an important factor in tumor infiltration and metastasis, making it possible to apply specific inhibitors and targeted drugs to impede HMGN5's pro-tumor effect and exert anti-tumor efficacy. It is envisaged that in the imminent future, the HMGN family could emerge as a novel focal point for clinical tumor therapy, conferring potential benefits to a broader spectrum of patients afflicted with malignant tumors. In recent years, there has been mounting evidence suggesting that the HMGN gene family assumes a substantial role in the onset and development of tumor [35]. Nevertheless, there exists a scarcity of studies that have effectively demonstrated the pivotal role of the HMGN members in the initiation and development of liver cancer.

In conclusion, HMGs have diverse roles in basic biology and the pathogenesis and progression of tumors. However, so far, there has been no study systematically evaluating the functional use of HMGs in HCC using bioinformatics methods. Hence, our objective is to ascertain the expression levels and clinical significance of HMGs in HCC. The aim of this endeavor is to furnish robust scientific evidence for liver cancer research, facilitating informed clinical decision-making and aiding in the risk management of liver cancer patients.

## Materials and methods

### HMG family expressions and data obtained

In this study, we employed R software (version 3.6.3) to analyze the HMGs expression levels in both tumor and paracancerous tissues from the LIHC TCGA cohort, as well as normal liver tissue from the GTEx database (https://genome.ucsc.edu/). The resulting visualizations were created using ggplot2 (version 3.3.3). The assessment of statistical significance for differential expression was conducted using the Wilcoxon test. Statistical significance was designated at P values below 0.05.

### The human protein atlas (HPA)

Immunohistochemistry staining images for HCC and normal tissues were acquired from the Human Protein Atlas (HPA) (https://www.proteinatlas.org/). The HPA employs transcriptome and proteomics methodologies to furnish a comprehensive collection of protein atlases, encompassing tissue, cell, and pathology details.

### Gene expression profiling interactive analysis (GEPIA)

Leveraging the capabilities of GEPIA (http://gepia.cancer-pku.cn/), we conducted gene expression analysis to discern disparities between LIHC and normal tissues. To achieve this, we employed analysis of variance (ANOVA) to generate scatter diagrams and box plots. Furthermore, GEPIA enabled the evaluation of the correlation between HMGs and clinical staging, utilizing the Pearson correlation coefficient as the statistical method.

### Survival analysis of HMGs members In HCC

We classified the HCC samples into two groups based on the HMGs expression levels. Overall survival (OS) was defined as the period from the initial diagnosis of HCC to either death or the last follow-up, whichever occurred first. Disease-specific survival (DSS) serves as an outcome measure, reflecting death specifically caused by the disease in question and indicating the clinical benefit associated with that particular disease. On the other hand, progression-free Interval

(PFI) represents the period during which the disease does not progress after treatment, and the outcome measure is the occurrence of disease progression or death. Hazard ratios (HR) along with their corresponding P-values were computed, and significance was attributed to P-values < 0.05.

## Receiver operating characteristic (ROC) curves of HMGs in HCC

In our investigation, we employed RNA-seq data in TCGA-LIHC FPKM format, subsequently converting it to TPM format and applying a log2 transformation. Through R software (×64 3.6.3), we conducted ROC curve analysis, specifically constructing ROC curves for imaging markers associated with Overall Survival (OS), and quantitatively assessing the areas under the ROC curves (AUC) utilizing the trapezoid rule. The analytical process was conducted utilizing the "pROC" package, while visualization was achieved using the "ggplot2" package.

## LinkedOmics database

We employed the LinkedOmics database to identify the most pertinent genes associated with each member of the HMG family. The heatmap and volcano plot depicted the top 50 genes that exhibited significant associations with the HMGs.

## Immune infiltration analysis

Immune cell recognition and infiltration within tumors play a critical role in identifying and eradicating cancer. For the purpose of immune infiltrates analysis, we utilized the "GSVA" package within the R software (version ×64 3.6.3), coupling it with the ssGSEA algorithm and leveraging RNA-seq data sourced from TCGA-LIHC in FPKM format. Furthermore, we conducted statistical evaluations using the Welch's t-test to compare subgroups of HMGs and immune infiltrates.

## Heat map correlational analysis

Spearman's correlation coefficient was conducted to assess the associations among the two High Mobility Group genes (HMGs), their most relevant genes, and immune checkpoints. Statistical analysis was conducted using R software (V3.6.3), and the results were visualized through plots. We created a heat map to visualize the top 50 genes that exhibited significant associations with HMGs. The nine immune checkpoints analyzed encompassed (PD-1) PDCD1, (PD-L1)CD274, (PD-L2)PDCD1LG2, CTLA4, LMTK3, LAG3, TIGIT, HAVCR2, and SIGLEC15. P-value below 0.05 were deemed statistically significant.

## Gene ontology(GO) and Kyoto encyclopedia of genes and genomes(KEGG) analysis

We conducted Gene Ontology (GO) and Kyoto Encyclopedia of Genes and Genomes (KEGG) enrichment analyses using the DAVID database (https://david.ncifcrf.gov/). Significantly enriched pathways were identified using statistical calculations provided by the DAVID database. The results of both GO and KEGG analyses were visualized applying the ggplot2 R package (version 3.6.3).

## Cell culture

Human normal hepatocyte (HL-7702) and HCC (Hep3B, Huh7, MHCC97-H, MHCC97-L and SK-Hep-1) cell lines were obtained from the Cell Bank of the Chinese Academy of Sciences (Shanghai, China). The HL-7702, MHCC97-L and SK-Hep-1cell lines were grown in

RPMI1640 (ThermoFisher Biochemical Products Beijing Corporation, China) medium, and the Huh7 and MHCC97-H cell lines were grown in DMEM medium (ThermoFisher Biochemical Products Beijing Corporation, China), while the Hep3B cell line was grown in MEM (including NEAA) medium (Servicebio Corporation, Wuhan, China). The culture medium comprised 10% fetal bovine serum (FBS) sourced from Procell Corporation (Wuhan, China), alongside 100 U/ml penicillin and 100 μg/ml streptomycin from NCM Biotech (Suzhou, China). The cells were cultured in a humidified atmosphere at 37˚C with a 5% $CO_2$ supplementation.

### Quantitative real-time polymerase chain reaction (qRT-PCR) analysis

We employed the RNA-easy™ Isolation Reagent from Vazyme Company (Nanjing, China) for total RNA extraction, following the manufacturer's guidelines. The extracted RNA was subsequently reverse transcribed into complementary DNA (cDNA) using another product from Vazyme Company, which named HiScript Q RT SuperMix for qPCR reagent Kit. Real-time PCR analysis was then conducted using the SYBR Green (Seven/Abcells, Beijing, China) detection method. Data represent the mean ± standard deviation (SD) of three independent experiments, each performed in triplicate (n = 3). The primer pairs utilized for the target genes in the qRT-PCR assay are provided in (Table 1).

### siRNA transfection

We seeded healthy MHCC97-H and Hep3B cells in a 6-well plate. After 24 hours, we performed cell transfection using the Lipofectamine 2000 reagent, following the manufacturer's instructions. Cells transfected with si-NC served as the control group, while cells transfected

**Table 1. Sequences of primer pairs for target genes used in the qRT-PCR.**

| Genes | Primer Sequences (from 5' to 3') |
|---|---|
| HMGA1 | Forward: GAAGTGCCAACACCTAAGAGACC |
|  | Reverse: GGTTTCCTTCCTGGAGTTGTGG |
| HMGA2 | Forward: GAAGCCACTGGAGAAAAACGGC |
|  | Reverse: GGCAGACTCTTGTGAGGATGTC |
| HMGB1 | Forward: CTTGTCGGGAGGAGCATAAGAA |
|  | Reverse: ATGGTCTTCCACCTCTCTGAGC |
| HMGB2 | Forward: GGTGAAATGTGGTCTGAGCAGTC |
|  | Reverse: CCTGCTTCACTTTTGCCCTTGG |
| HMGB3 | Forward: TTTTCCAAGAAGTGCTCTGAGA |
|  | Reverse: TTCTTCTTCTTGCC TCCCTTAG |
| HMGN1 | Forward: AAAGTGCAAACAAAAGGGAAAAGG |
|  | Reverse: ATCAGACTTGGCTTCTTTCTCTCC |
| HMGN2 | Forward: AAACCTGCTCCTCCAAAGCCAG |
|  | Reverse: CTTGCCAGCATCAGCTTTTCCC |
| HMGN3 | Forward: TCTCCAGAGAATACAGAGGGCAAA |
|  | Reverse: GCTCCAGGTTCTTTCTTAGCAGAT |
| HMGN4 | Forward: CACAGAGGAGATCAGCTCGGTT |
|  | Reverse: GGTTGTTCCCATCCTTTCCAGC |
| HMGN5 | Forward: AAGCAGTTGCTGAAACCAAGC |
|  | Reverse: TCCCTTTTCTGTGGCATCTTC |
| GAPDH | Forward: TGACCTGCCGTCTAGAAAAACCT |
|  | Reverse: GCTGTTGAAGTCAGAGGAGACCA |

with si-HMGA2 formed the experimental group. Transfection efficiency was validated using quantitative PCR (qPCR). The small interfering RNA (siRNA) sequence used for HMGA2 knockdown was as follows: siRNA-HMGA2: (sense, 5′-GGAAGAACGCGGUGUGUAACA-3′; antisense, 5′-UUACACACCGCGUUCUUCCUA-3'); siRNA-NC: (sense,5'-UUCUCCGAACGU GUCACGUTT-3′; antisense, 5-ACGUGACACGUUCGGAGAATT-3′).

## Western blotting

A total of 1×10^6 cells were treated for total protein extraction using RIPA lysis buffer (G2002, Servicebio, China) supplemented with a protease inhibitor cocktail (G2006, Servicebio, China) and a phosphatase inhibitor mixture (P1260, Servicebio, China). The tissues were lysed, and the lysate was centrifuged at 12,000 g for 15 minutes at 4˚C to obtain the total protein. Protein concentrations were determined using the BCA Protein Assay Kit (Biosharp, China). Western blotting was then performed using sodium dodecyl sulfate-polyacrylamide gel electrophoresis (SDS-PAGE). The blots were visualized with the Tanon 5200 chemiluminescence system, and the densities were analyzed using ImageJ software. HMGA2 polyclonal antibodies (1:10,000; Cat No. 20795-1-AP; Proteintech Group, Inc.) and GAPDH monoclonal antibodies (1:20,000; Cat No. 10494-1-AP; Proteintech Group, Inc.) were used as primary antibodies.

## CCK-8 assay

We initially plated healthy MHCC97-H and Hep3B cells in a 96-well plate at a density of 2000 cells per well. After a 24-hour adherence period, we divided the cells into two groups: one received si-NC transfection, while the other received si-KIF3C transfection. At 0, 24, 48, and 72 hours post-transfection, we added 10 μL of CCK-8 solution to each well and incubated the cells at 37˚C for 1 hour. Finally, we measured the absorbance at 450 nm using a microplate reader.

## Wound-healing and transwell experiments

MHCC97-H and Hep3B cells were initially seeded in a 6-well plate and transfected upon reaching 60–70% confluence. After transfection, when the cells reached 100% confluence, a scratch was created along the axis of the cell monolayer using a 10 μL pipette tip. The detached cells were washed three times with PBS, and serum-free basal culture medium was added. The plate was then returned to the incubator for further incubation. Wound healing was observed under an inverted microscope at 0 and 48 hours after creating the scratch.

## Transwell assays

We seeded healthy MHCC97-H and Hep3B cells in a 6-well plate, and after 24 hours, we harvested and counted the cells. A cell suspension containing 50,000 cells in 200 μL of serum-free basic culture medium was prepared and loaded into the Transwell chamber. Subsequently, 500 μL of culture medium containing 20% FBS was added beneath the chamber. The Transwell insert was then placed in a cell culture incubator. After a 24-hour incubation period, the fluid in the upper chamber was aspirated, and the cells were fixed with 4% paraformaldehyde for 30 minutes. Following fixation, the cells were stained with crystal violet for 20 minutes. The cells were observed, photographed, and counted under a microscope. For the invasion assay, 5×10^4 cells were seeded in Transwell chambers pre-coated with Matrigel. The chamber was loaded with serum-free medium, while medium containing 20% FBS was placed beneath it. After 24 hours, the cells that had traversed the Matrigel barrier were counted under a microscope.

## Results

### The HMGs expressions in human cancers

The mRNA expression levels of individual subfamilies within the HMGs were evaluated in both cancerous and para-cancerous tissues using the TCGA database. The results revealed that in the majority of cancers, all HMGs were elevated in comparison to para-cancerous tissues, as shown in (Fig 1).

### The mRNA and protein expression levels of HMGs in HCC

In order to assess the mRNA expression levels of HMG members in HCC, we conducted an analysis of transcriptomic data retrieved from the TCGA database, which included 374 HCC tissues and 50 para-cancerous tissues. Our findings revealed all HMGs exhibited a significant increase compared to para-cancerous tissues (p<0.05) (Fig 2A and 2B).

The protein levels of HMGs in HCC were examined through immunohistochemistry staining images obtained from HPA, as depicted in (Fig 3). With the exception of HMGN4 due to lack of available data, the protein levels of the other HMGs were found to be elevated in HCC tissues as opposed to para-cancerous tissues.

### Validation of mRNA expression levels in HCC cell lines

Following this, a qRT-PCR analysis was conducted to validate HMGs expresions in HCC cell lines, as depicted in (Fig 4). The findings revealed that, in comparison to normal liver cells (HL-7702), the HMGA1 was notably elevated in various HCC cell lines (Hep3B, Huh7, MHCC97-H, MHCC97-L), with the exception of SK-Hep-1. HMGA2 exhibited elevated expression across all liver cancer cell lines. HMGB1 displayed high expression exclusively in Huh-7. HMGB2 demonstrated heightened expression in all liver cancer cell lines except MHCC-97H. As for HMGB3, it exhibited high expression in MHCC-97H and 97L. HMGN1 exhibited elevated expression levels in Huh7 cell line. HMGN2 demonstrated high expression in Huh7 and SK-Hep-1 cells, but lower expression in the remaining cell lines. HMGN3 was only downregulated in MHCC-97H, while it was highly elevated in other liver cancer cells. The expression pattern of HMGN4 differed from other members, it was highly expressed in MHCC97-L, and SK-Hep-1, while being lowly expressed in other cell lines. HMGN5 showed high expression in the majority of HCC cell lines, consistent with our predictions.

### Correlation between clinical pathological characteristics and HMGs

We subsequently conducted an examination of the correlation between differential expression of HMGs and the clinical pathological stage of HCC patients. This analysis, aimed at elucidating the correlations between tumor stage and HMGs, utilized data from the GEPIA (Fig 5A). The elevated mRNA expression of HMGA1/A2/B1/B2/N2/N3 in tumor tissues exhibited correlation with the cancer stages of HCC patients. Furthermore, univariate logistic regression analysis revealed that HMGA2 was associated with Gender (Male vs. Female, p < 0.01),T stage (Pathologic T1 vs. T2, p = 0.006), pathologic stage (Stage II&Stage III&Stage IV vs. Stage I, p = 0.019), AFP(ng/ml) > 400 vs. < = 400, p<0.001) and Vascular invasion (Yes vs. No, p = 0.039) (Table 2). Therefore, our findings imply that high HMGA2is associated with unfavourable clinicopathological features.

In addition to assessing the tumor prognosis, we employed survival analysis to examine the influence of HMGs on the prognosis of HCC (Fig 5B). The group with lower HMGA1 expression exhibited significantly improved overall survival (OS) and progression-free interval (PFI), while the group with higher HMGN2 expression displayed significantly worse overall survival

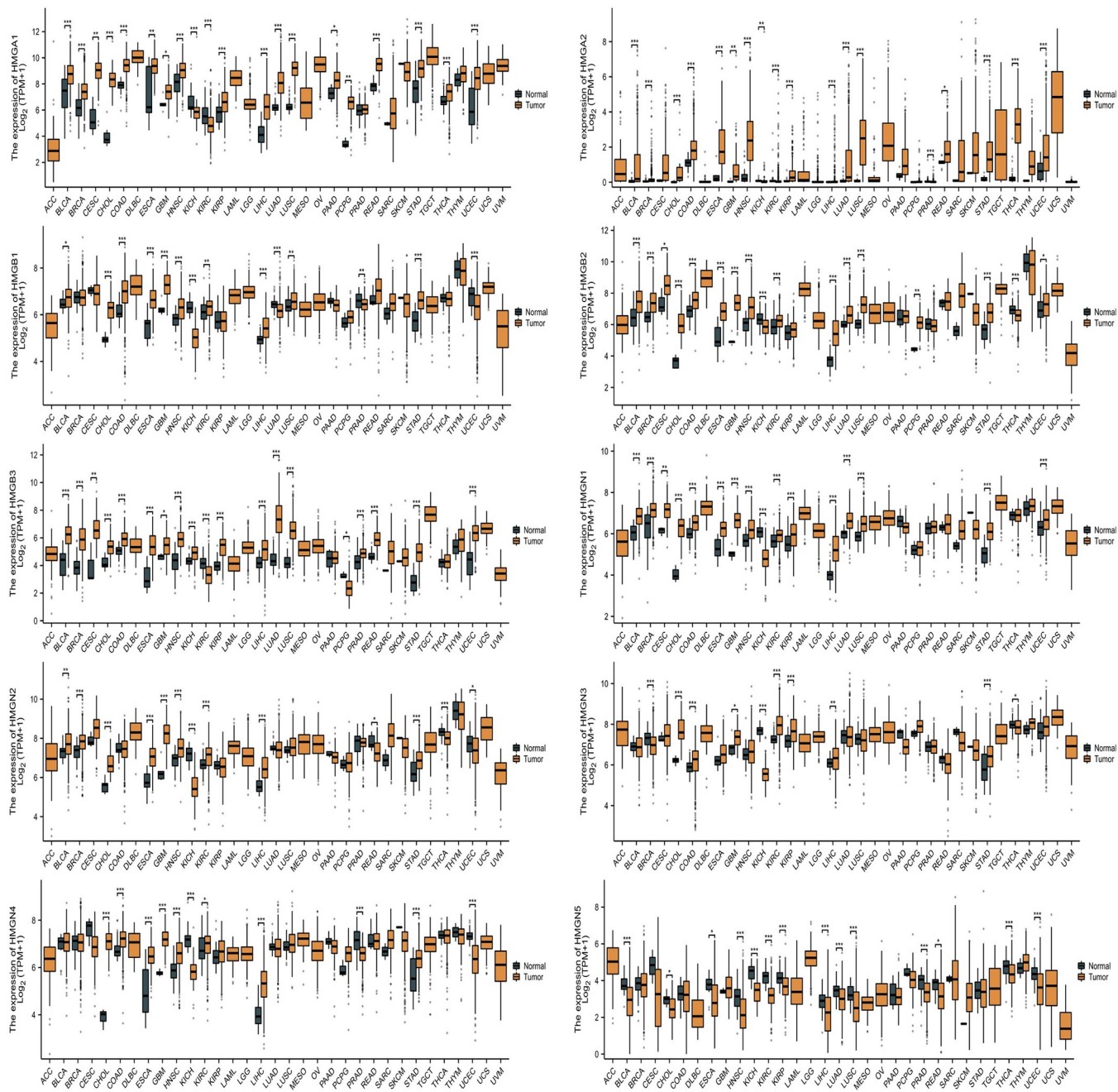

**Fig 1. mRNA expression levels of HMGs in human cancers.** The numbers of Normal group (N) and Tumor group (T) in different types of cancer were: ACC (Adrenocortical Carcinoma): N: 0,T: 79; BLCA (Bladder Urothelial Carcinoma): N: 19, T: 412; BRCA (Breast Invasive Carcinoma): N: 113, T: 1113; CESC (Cervical Squamous Cell Carcinoma and Endocervical Adenocarcinoma): N: 3, T: 306; CHOL (Cholangio Carcinoma): N: 9, T: 35; COAD (Colon Adenocarcinoma): N: 41, T: 480; DLBC (Lymphoid Neoplasm Diffuse Large B-cell Lymphoma): N: 0, T: 48; ESCA (Esophageal Carcinoma): N: 11, T: 163; GBM (Glioblastoma Multiforme): N: 5, T: 169; HNSC (Head and Neck Squamous Cell Carcinoma): N: 44, T: 504; KICH (Kidney Chromophobe): N: 25, T: 65; KIRC (Kidney Renal Clear Cell Carcinoma): N: 72, T: 541; KIRP (Kidney Renal Papillary Cell Carcinoma): N: 32, T: 291; LAML (Acute Myeloid Leukemia): N: 0,T: 150; LGG (Brain Lower Grade Glioma): N: 0,T: 532; LIHC (Liver Hepatocellular Carcinoma): N: 50, T: 374; LUAD (Lung Adenocarcinoma): N: 59, T: 539; LUSC (Lung Squamous Cell Carcinoma): N: 49, T: 502; MESO (Mesothelioma): N: 0, T: 87; OV (Ovarian Serous Cystadenocarcinoma): N: 0, T: 381; PAAD (Pancreatic Adenocarcinoma): N: 4, T: 179; PCPG (Pheochromocytoma and Paraganglioma): N: 3, T: 184; PRAD (Prostate Adenocarcinoma): N: 52, T: 501; READ (Rectum Adenocarcinoma): N: 10, T: 167; SARC (Sarcoma): N: 2, T: 263; SKCM (Skin Cutaneous Melanoma): N: 1, T: 472; STAD (Stomach Adenocarcinoma): N: 32, T: 375; TGCT (Testicular Germ Cell Tumors): N: 0, T: 156; THCA (Thyroid Carcinoma): N: 59, T: 512; THYM (Thymoma): N: 2, T: 120; UCEC (Uterine Corpus Endometrial Carcinoma): N: 35, T:554; UCS (Uterine Carcinosarcoma): N: 0, T: 57; UVM (Uveal Melanoma): N: 0, T: 80. (*p< 0.05; **p<0.01; ***p<0.001; ns, no statistically significant difference).

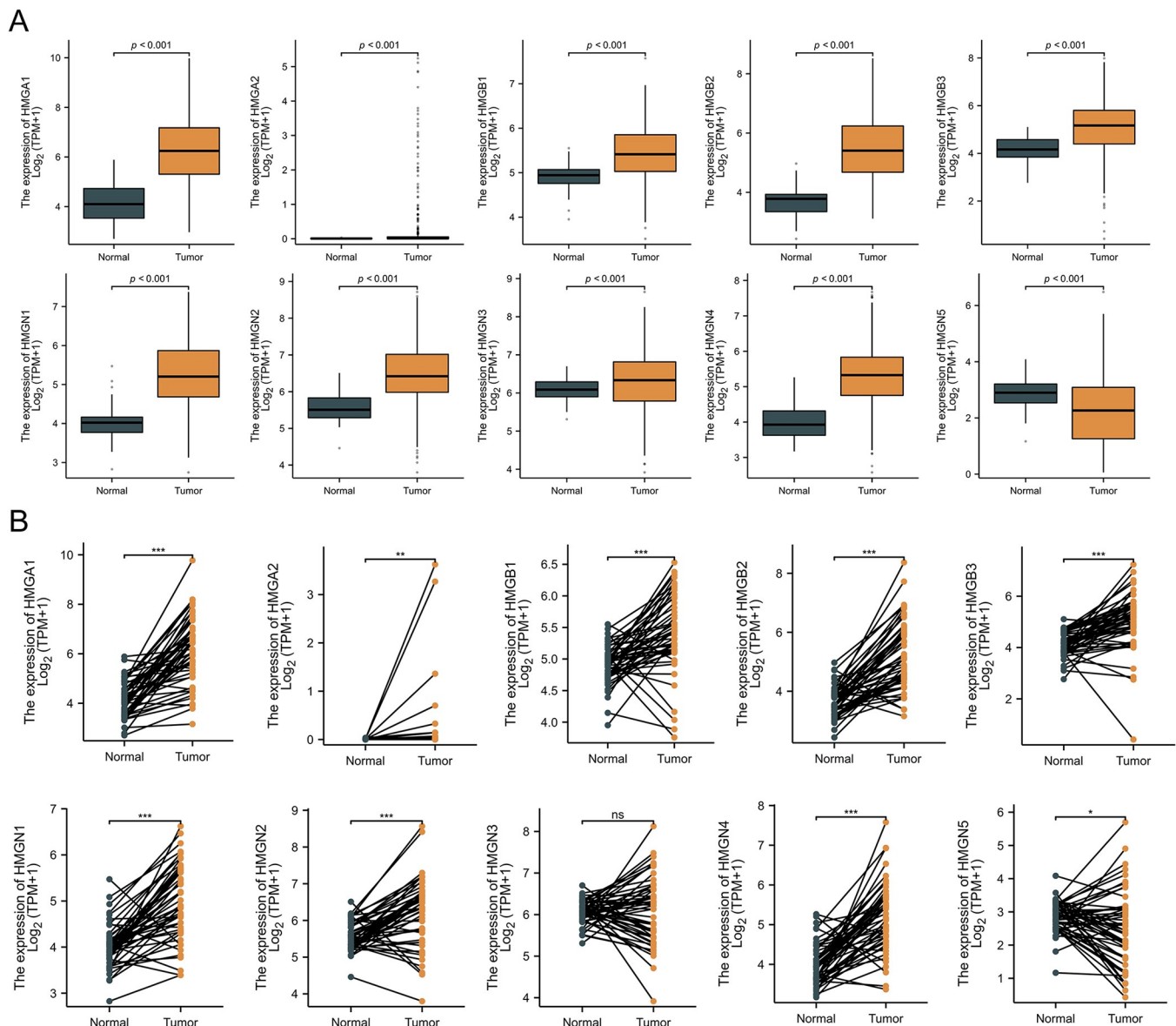

**Fig 2. Expression patterns of HMGs in LIHC.** (A). The mRNA expression levels of HMGs in LIHC, Normal = 50,Tumor = 374. (B) The expression of HMGs in tumors and adjacent normal tissues in TCGA, Normal = 50, Tumor = 50. (*p< 0.05; **p<0.01; ***p<0.001; ns, no statistically significant difference).

(OS) and disease-specific survival (DSS) (p < 0.05). The disease-specific survival (DSS) of the group with high HMGA2 expression was shorter compared to the low expression group, and the progression-free interval (PFI) of the group with high expression of HMGN3 was also shorter in comparation with the low expression group (p < 0.05). We additionally observed significant correlations between high expressions of HMGB2, HMGN1, and HMGN4 with the prognoses of HCC patients (p < 0.05). However, the expressions of HMGB1, HMGB3, and HMGN5 did not exhibit significant effects on the prognosis of HCC patients. Our study also conducted a comprehensive Cox regression analysis to explore the prognostic significance of various HMG family genes in HCC, assessing their impact on OS, DSS, and PFI.We found that HMGA1, HMGA2, HMGB2, HMGN1, HMGN2, and HMGN4 emerged as significant genes across multiple prognostic indicators (OS, DSS, PFI), suggesting their potential

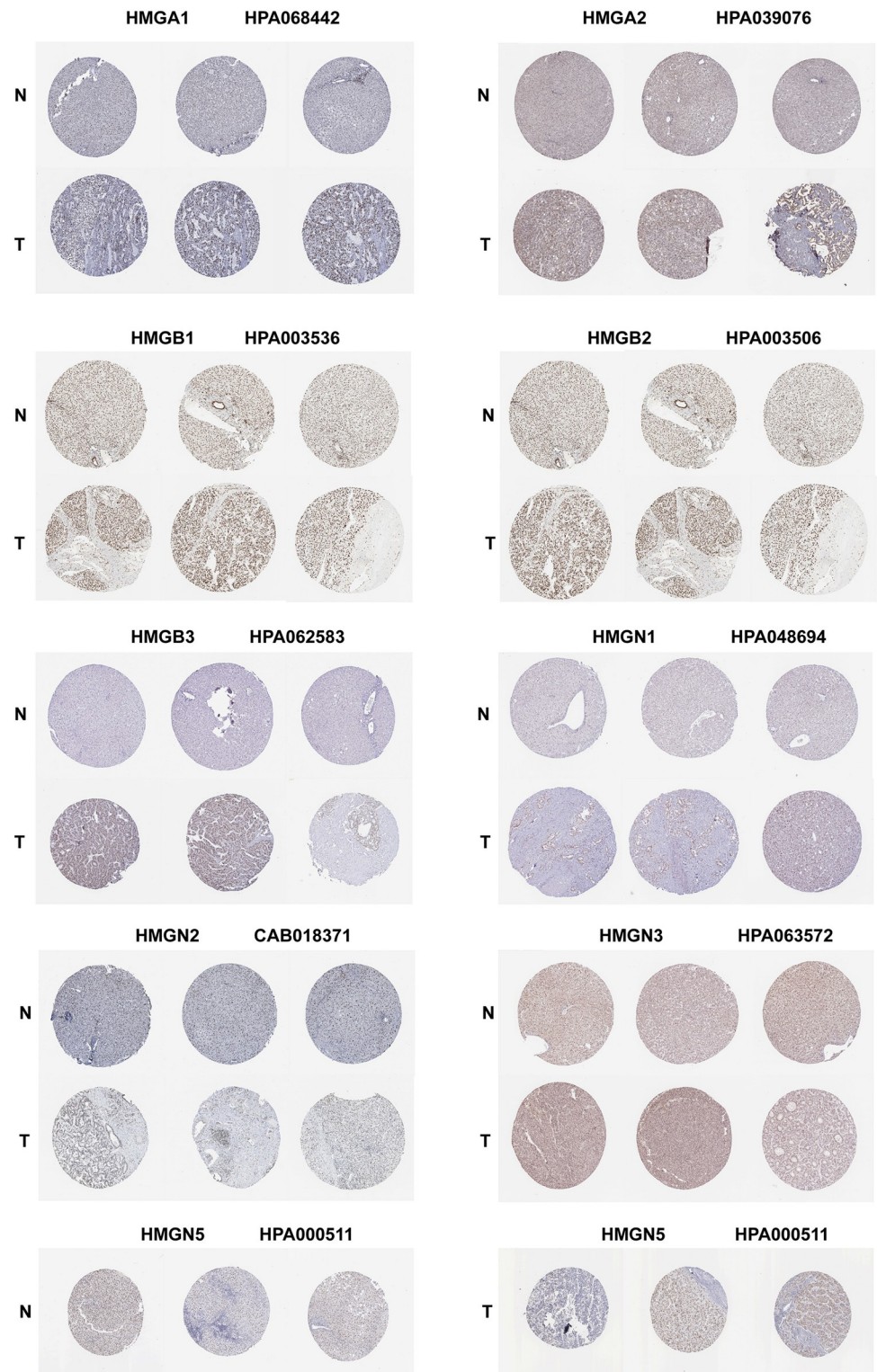

**Fig 3. Immunohistochemical analysis of HMGs in the HPA-derived 3 normal, and LIHC samples.**

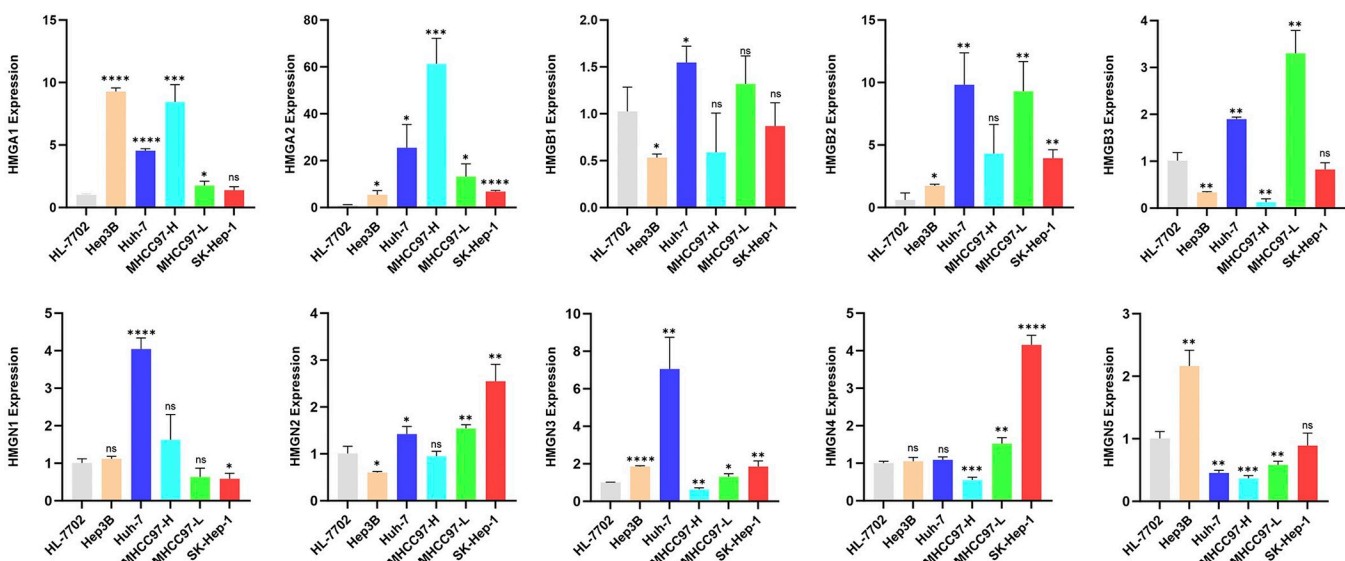

**Fig 4. mRNA expression of HMGs in four liver cancer cell lines.** The relative expression levels of HMGs in HCC cell lines (HL-7702, Hep3B, Huh7, MHCC97-H, MHCC97-L and SK-Hep-1) were detected by qRT-PCR (*p< 0.05; **p<0.01; ***p<0.001; ns, no statistically significant difference).

importance in the prognosis of HCC (S1–S3 Tables). However, their independent prognostic value is less clear when accounting for other clinicopathological factors. These genes, in conjunction with clinical features, could contribute to a more nuanced understanding of HCC prognosis.

## Exploration of potential diagnostic markers for HCC

We proceeded to evaluate the potential of HMGs as biomarkers for HCC. The ROC curves analysis demonstrated that HMGA1/B2/N1/N4 effectively distinguish HCC from normal samples, yielding AUC values of HMGA1(AUC = 0.913), HMGB2(AUC = 0.939), HMGN1 (AUC = 0.905), HMGN4(AUC = 0.904). HMGB1 and HMGB3 show comparable values, both with AUCs of 0.782 and 0.788 respectively. While HMGN2 presents moderate accuracy, achieving an AUC of 0.839. The potential of HMGA2, HMGN3, and HMGN5 in distinguishing HCC from normal tissue remains relatively low. ROC curves analysis revealed an AUC of 0.648 for HMGA2, 0.627 for HMGN3 and 0.676 for HMGN5 (Fig 6A). To validate the effectiveness of the aforementioned HMGs genes in the diagnosis of HCC, ROC analysis was conducted based on the GSE84402 and GSE76427 datasets. The analysis revealed that the AUC values for HMGA1, HMGB2, HMGB3, HMGN1, HMGN2, and HMGN4 were all greater than 0.7 (S1A and S1B Fig), demonstrating that these six HMGs genes have good sensitivity and specificity in diagnosing HCC. Subsequently, ROC curves analysis ware conducted to evaluate the prospective utility of HMGs by the clinical stage, confirming consistent findings (Fig 6B). Hence, HMGA1/B2/N1/N4 could emerge as pivotal diagnostic biomarkers for patients with HCC. Notably, time-dependent ROC analysis indicates that only HMGA1 boasts AUC values surpassing 0.6 for predicting the 1–5 year survival rate of patients with HCC (Fig 6C).

## Significant genes associated with HMGs

The linkedOmics database was employed to investigate the significant genes correlated with the HMGs. The volcano plot in (Fig 7) and the heatmap plot in (Fig 8) display the top 50 correlated genes. We observed notable correlations between HMGAs and various genes. For

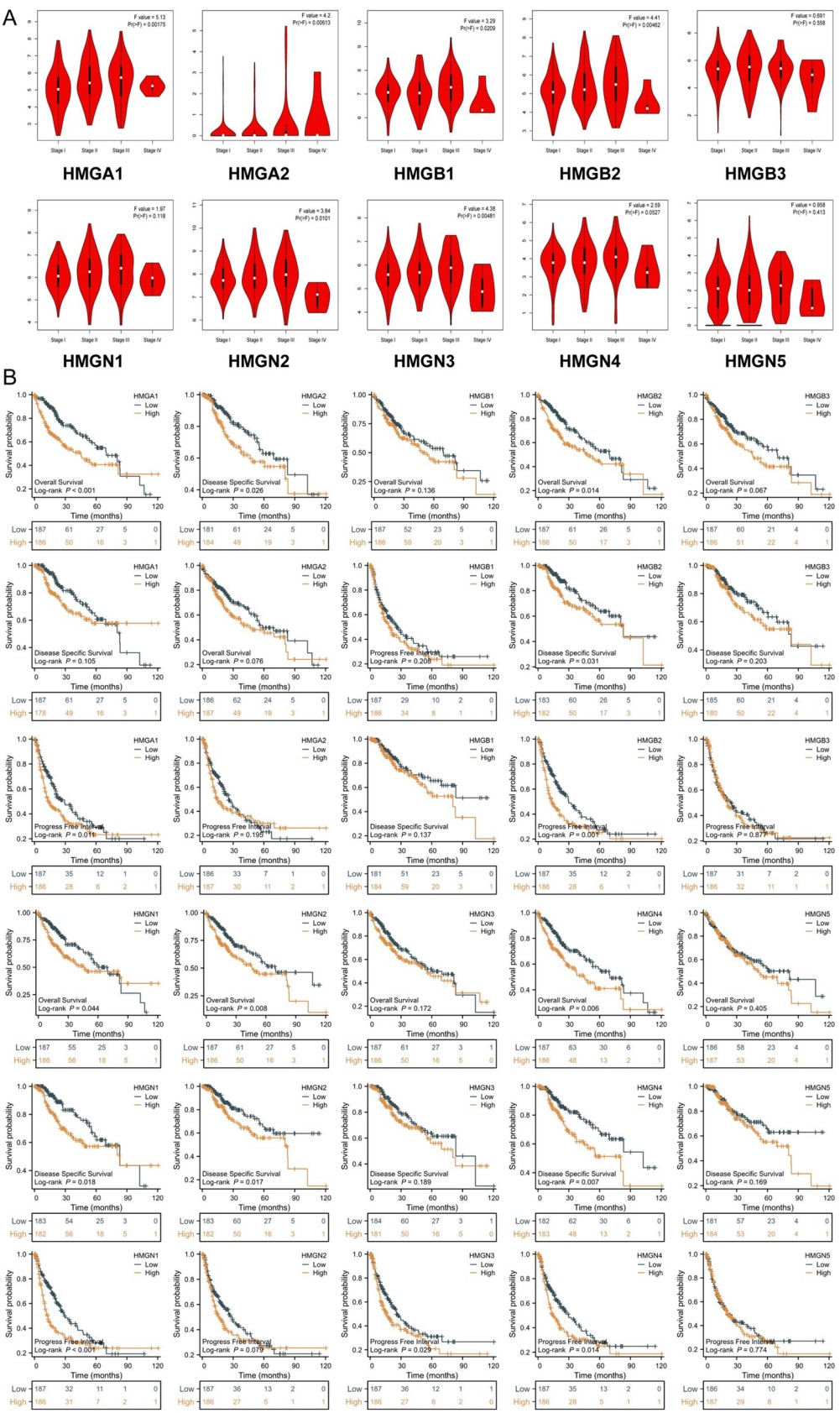

**Fig 5. Effects of HMGs expressions on prognosis in LIHC.** (A) Association between HMGs and tumor stage in HCC; (B) Effect of HMGs on OS, DSS and PFI in HCC.

HMGA1, negatively correlated genes encompassed DNAJC25, GLYALT1, and CYP4V2, while positively correlated genes included MFSD2B, EHMT2, and CLIC1. HMGA2 displayed negative correlations with SERPINC1, HSD17B6, and HAGH, and positive correlations with IGDCC4, IGF2BP2, and KCTD17. HMGB1 was negatively associated with TBC1D9B, ATP6V0A1, and CTSD, and positively associated with EXOSC8, N4BPIL2, and POLR1D. HMGB2 demonstrated negative correlations with MLYCD, NDUFAF1, and KCTD21, while positive correlations were observed with CCNA2, PLK4, and NUSAP1. HMGB3's negative correlations included ZNRF3, LGR5, and RNF43, and its positive correlations involved CD99L2, LOC145837, and SDS. HMGN1's negative correlations incorporated AR, AOX1, and KAT2B, whereas its positive correlations encompassed U2AF1, C21orf45, and NRM. HMGN2 was negatively associated with UBE2H, HERC4, and TGOLN2, and positively associated with DNAJC8, STMN1, and C15orf21. HMGN3 displayed negative correlations with NCOA2, OPLAH, and TTPA, and positive correlations with LYRM2, HDDC2, and C6orf162. HMGN4's negative correlations included ECHS1, SLC22A3, and RPB4, and positive correlations encompassed BTN2A1, CD2AP, and HCG18. Finally, HMGN5 exhibited negative correlations with PDE9A, LZTS2, and DNMT3A, and positive correlations involved ATP11C, NAT2, and ACADL.

## Association between immune infiltration and HMGs

Tumor occurrence is intricately linked with the immune system, and the pivotal role of tumor-infiltrating lymphocytes in tumor development significantly influences the treatment outcomes and prognosis of HCC patients. Hence, we conducted an investigation to determine the potential correlation between HMGs and the extent of immune infiltration in HCC (Fig 9). The findings revealed a positive association between the mRNA expression of HMGA1 and aDC cells, while demonstrating a negative association with TReg and Eosinophils. The mRNA

**Table 2. Relationship between HMGA2 expression and clinical characteristics using logistic regression.**

| Characteristics | Total (N) | OR (95% CI) | P value |
|---|---|---|---|
| Gender (Male vs. Female) | 374 | 0.539 (0.347–0.838) | **0.006** |
| Age (> 60 vs. < = 60) | 373 | 0.686 (0.456–1.032) | 0.070 |
| Pathologic T stage (T1 vs. T2) | 278 | 0.549 (0.333–0.905) | **0.019** |
| Pathologic N stage (N1 vs. N0) | 258 | 2.953 (0.303–28.768) | 0.351 |
| Pathologic M stage (M1 vs. M0) | 272 | 0.942 (0.131–6.786) | 0.953 |
| Pathologic stage (Stage II&Stage III&Stage IV vs. Stage I) | 350 | 2.250 (1.467–3.451) | **< 0.001** |
| Histologic grade (G2&G3&G4 vs. G1) | 369 | 1.578 (0.881–2.828) | 0.125 |
| Weight (> 70 vs. < = 70) | 346 | 0.687 (0.449–1.050) | 0.083 |
| Height (> = 170 vs. < 170) | 341 | 0.770 (0.500–1.187) | 0.236 |
| BMI (> 25 vs. < = 25) | 337 | 0.724 (0.472–1.112) | 0.141 |
| Adjacent hepatic tissue inflammation (Mild&Severe vs. None) | 237 | 0.776 (0.465–1.295) | 0.331 |
| AFP(ng/ml) (> 400 vs. < = 400) | 280 | 4.765 (2.567–8.843) | **< 0.001** |
| Albumin(g/dl) (> = 3.5 vs. < 3.5) | 300 | 1.170 (0.679–2.014) | 0.572 |
| Child-Pugh grade (B&C vs. A) | 241 | 0.974 (0.399–2.375) | 0.953 |
| Vascular invasion (Yes vs. No) | 318 | 1.632 (1.025–2.600) | **0.039** |
| Prothrombin time (> 4 vs. < = 4) | 297 | 0.876 (0.531–1.443) | 0.602 |

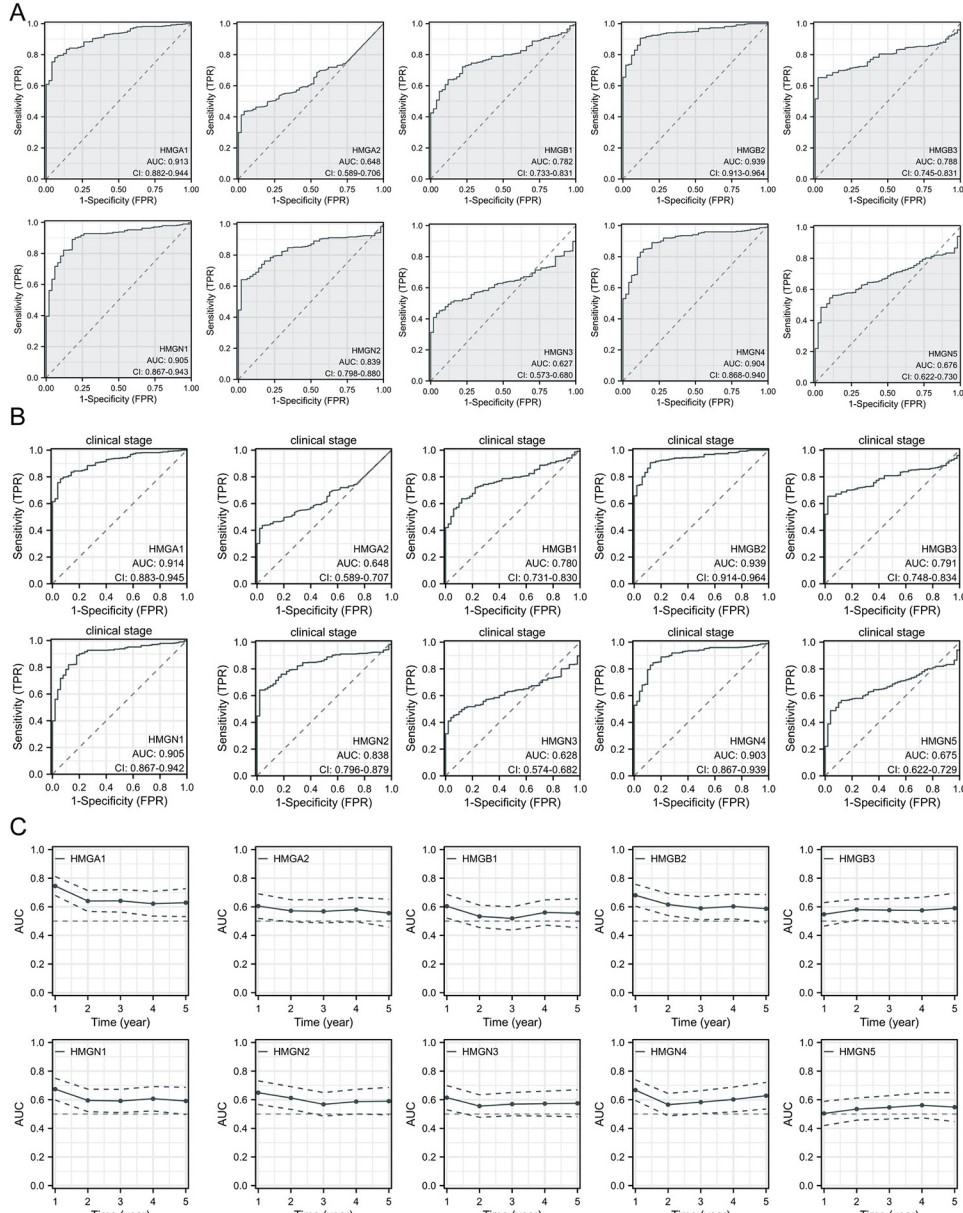

**Fig 6. The ROC curves were performed in LIHC.** (A) Diagnostic ROC curves were constructed to differentiate between HCC tissues and normal tissues based on the expression levels ofHMGs; (B) ROC curves with clinical stages in HCC; (C)Time-dependent survival ROC curves were generated to predict the 1- to 5-year survival rates of HCC patients using the expression levels of HMGs.

expression of HMGA2 exhibited a positive correlation with Microphages and CD56 bright cells, while displaying a negative correlation with Th17 cells. Levels of immune infiltrates exhibited similarity among HMGB1/N1/N2. Their respective mRNA expressions displayed a positive correlation with the Th2 cell and T helper cell, while demonstrating a negative correlation with pDCs. The mRNA expression of HMGB2 exhibited a positive correlation with Th2 cells, while showing a negative correlations with mast cells, Th17 cells, and NK cells. Furthermore, the HMGB3 expression displayed a positive correlation with Th2 cells, mast cells, and pDC cells. HMGN3 exhibited a positive correlation with TFH cells, while displaying a negative

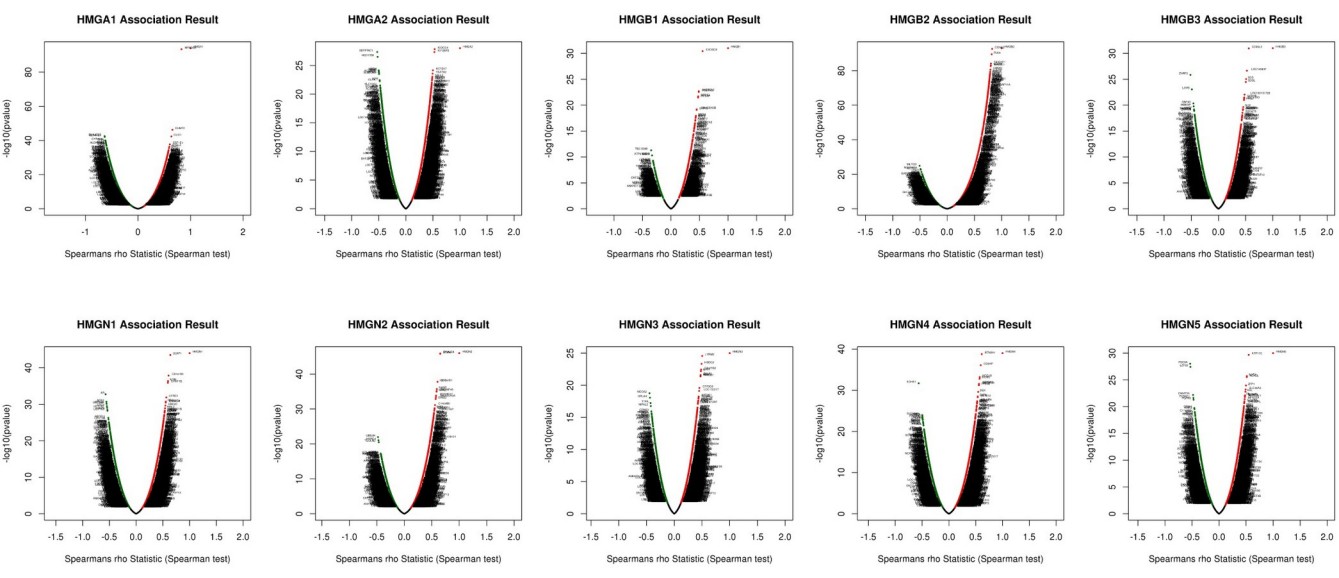

**Fig 7. Volcano plot of top 50 correlated genes to HMGs.**

correlation with Th17 cells. In contrast, HMGN4 demonstrated a positive correlation with T helper cells and Tcm cells, while significantly correlated negatively with pDC cells. HMGN5 showed a positive correlation with T helper cells and Neutrophils, while exhibiting a significant negative correlation with pDC cells. In summary, these findings suggest a strong association between HMGs and immune infiltrates in HCC.

## Relationship between immune checkpoints and HMGs in HCC

Given the potential oncogenic role of HMGs in HCC, we investigated their association with nine immune checkpoints, as illustrated in (Fig 10). Consequently, our investigation revealed a positive relationship between the HMGA1 expression level and the following immune checkpoints:PD-1, CTLA4, LMTK3, LAG3, TIGIT, and HAVCR2. Furthermore, our analysis demonstrated that the expression level of HMGA2 positively correlated with the immune checkpoints: PD-1, PD-L1, PD-L2, CTLA4, LMTK3, LAG3, TIGIT, and HAVCR2 in the context of HCC. HMGB1 exhibited a positive relationship with PD-1, PD-L1, PD-L2, CTLA4, LAG3, TIGIT, HAVCR2, and SIGLEC15. Similarly, HMGB2 displayed a notably strong and significant association with PD-1, PD-L1, PD-L2, CTLA4, LMTK3, LAG3, TIGIT, and HAVCR2 in the context of HCC. HMGB3 exhibited a robust positive correlation with PD-L2, LMTK3, and HAVCR2. HMGN1, HMGN2, and HMGN4 were correlated positively with all immune checkpoints. HMGN3 exhibited a positive association with PD-1, PD-L1, PD-L2, CTLA4, LAG3, TIGIT, and HAVCR2 in HCC. Additionally, HMGN5 demonstrated a significant positive association with PD-1, PD-L1, and PD-L2 in HCC. These findings suggest that the oncogenic processes of HCC, mediated by HMGs, may involve both tumor immune escape and antitumor immunity.

## GO and KEGG functional analysis

The HMGs and the aforementioned correlated genes underwent GO and KEGG enrichment analyses in the DAVID database. We showed top five processes in (Figs 11 and 12). Remarkably, the BPs were related to nuclear division, chromosome segregation and organelle fission (GO:0000280, 0007059, 0048285); and five MF processes associated with DNA activity

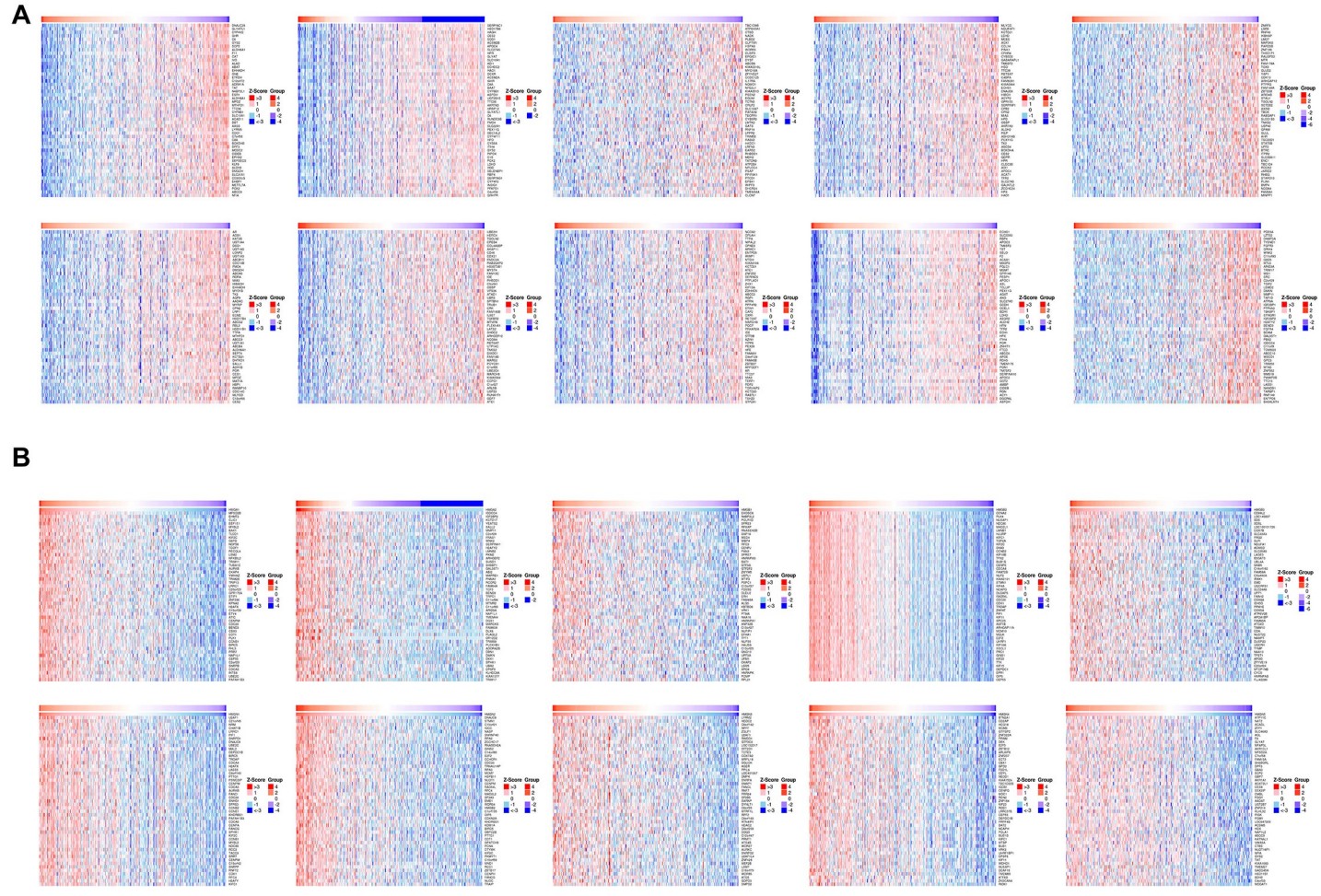

**Fig 8. Heatmap plot of top 50 associated genes of HMGs.** (A) negatively and (B) positively.

(GO:0017116, 0008094, 0140097, 0003697, 0003678), these findings strongly indicate the substantial relevance of HMGs to the cell cycle. This observation is supported by the KEGG analysis, which revealed that the cell cycle pathway ranked among the top five enriched pathways (Fig 11D).

## Knockdown of HMGA2 suppresses the malignant phenotype of HCC In vitro

Western blotting was then employed to examine HMGA2 protein expression in HCC cell lines. The results showed that HMGA2 expression was higher in the HCC cell lines Hep-3B and MHCC-97H compared to the HL-7702 cell line (Fig 12A). To investigate the influence of HMGA2 on the progression of liver cancer, we suppressed HMGA2 expression in liver cancer cells by transfecting them with si-HMGA2. Western blot results showed that HMGA2 protein expression was also significantly reduced in the si-HMGA2 group compared to the NC group (Fig 12B and 12C).The downregulation of HMGA2 in MHCC97-H and Hep3B cells was confirmed post-transfection with si-HMGA2, compared to the control group transfected with si-NC (Fig 12D). We then performed CCK-8 assays to evaluate the effect of HMGA2 on cell proliferation. The optical density (OD) values for MHCC97-H and Hep3B cells transfected with

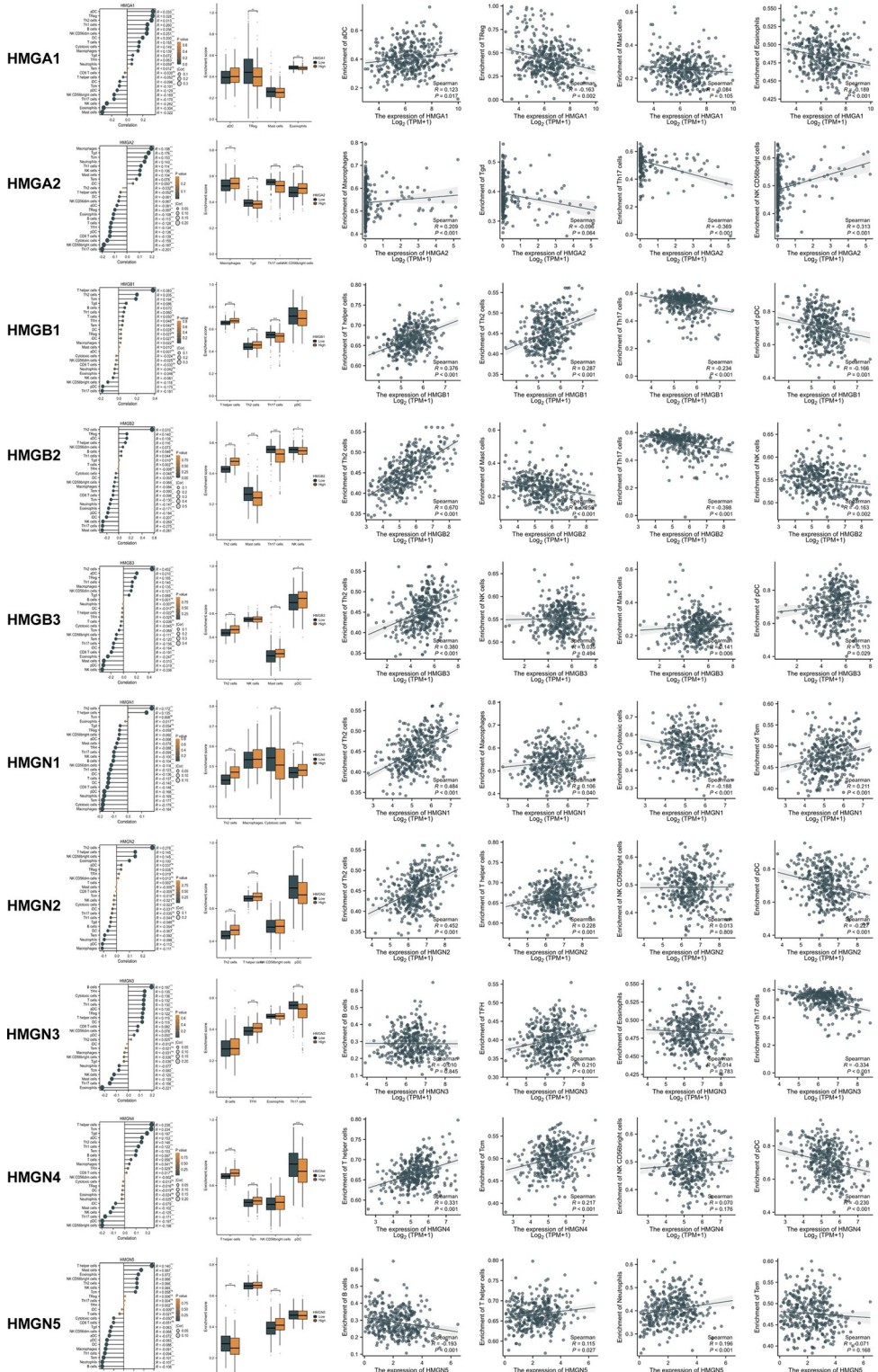

**Fig 9. The association between immune cell infiltration levels and HMGs was examined.** Subgroup comparison plots displayed the initial two cells exhibiting positive and negative correlations with HMGs. Correlation scatter plots visualized the robustness of the relationship between the four cells and HMGs in the subgroup comparison plots, determined through Spearman correlation analysis (*p < 0.05; **p < 0.01; ***p < 0.001).

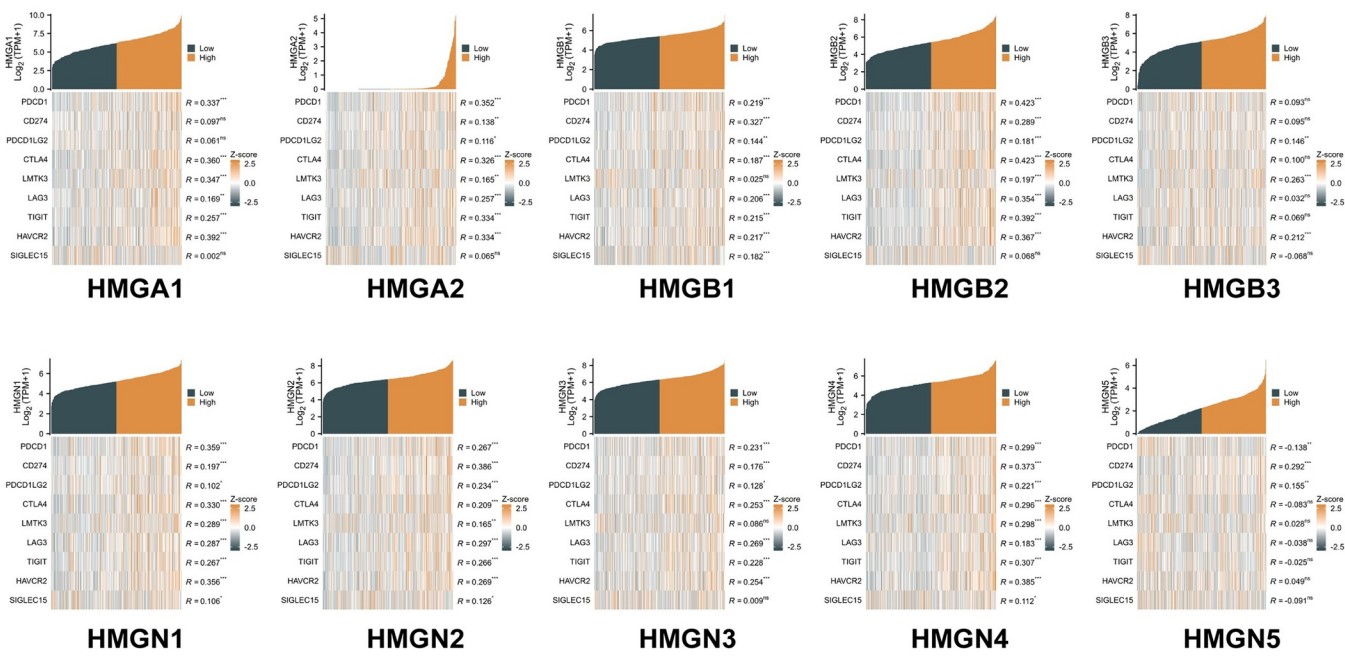

**Fig 10. Relevance analysis of immune checkpoint-related genes and HMGs in HCC (*p< 0.05; **p<0.01; ***p<0.001).**

si-HMGA2 showed a significant decrease compared to cells transfected with si-NC. This indicates a marked difference, as illustrated in (Fig 12E). Next, we examined the effect of HMGA2 knockout on tumor cell migration using wound healing assay. The results showed a markedly reduced number of migratory cells in the si-HMGA2 group compared to MHCC97-H and Hep3B cells transfected with si-NC (Fig 12F and 12G). This highlights the role of HMGA2 in enhancing the migration capabilities of liver cancer cells. In the Transwell migration and invasion assays, the si-HMGA2 group exhibited a significant decrease in migration and invasion ability compared to the si-NC group (Fig 12H–12K). These results collectively suggest that HMGA2 significantly enhances the invasion capacity of liver cancer cells.

## Discussion

We utilized various publicly available databases to provide the initial comprehensive and thoroughly analysis of HMGs in the context of HCC. Notably, all HMGs exhibited heightened expressions in HCC tissues in comparison with normal liver tissues, with the exception of HMGN5. We observed that HCC patients demonstrating elevated expressions of HMGA1/A2/N1/N3/N4 displayed shorter overall survival (OS) periods compared to those with lower expressions. This finding implies their potential utility as prognostic indicators in HCC. Furthermore, HCC patients showcasing elevated levels of HMGA1/A2/N1/N3/N4 expressions exhibited diminished progression-free intervals (PFI) relative to those with lower expression levels. This observation suggests the plausible candidacy of these markers as therapeutic targets for HCC treatment. We identified several genes that are significantly correlated with the expression of HMG family members in HCC. These correlations suggest potential interactions between these genes and HMGs that may play critical roles in HCC progression. For instance, genes such as MFSD2B, IGF2BP2, and CCNA2, which were positively correlated with HMGA1 and HMGA2, are known to be involved in cellular processes like metabolism, RNA binding, and cell cycle regulation, respectively. These functions align closely with the known roles of HMGs in chromatin remodeling and gene transcription regulation [36].

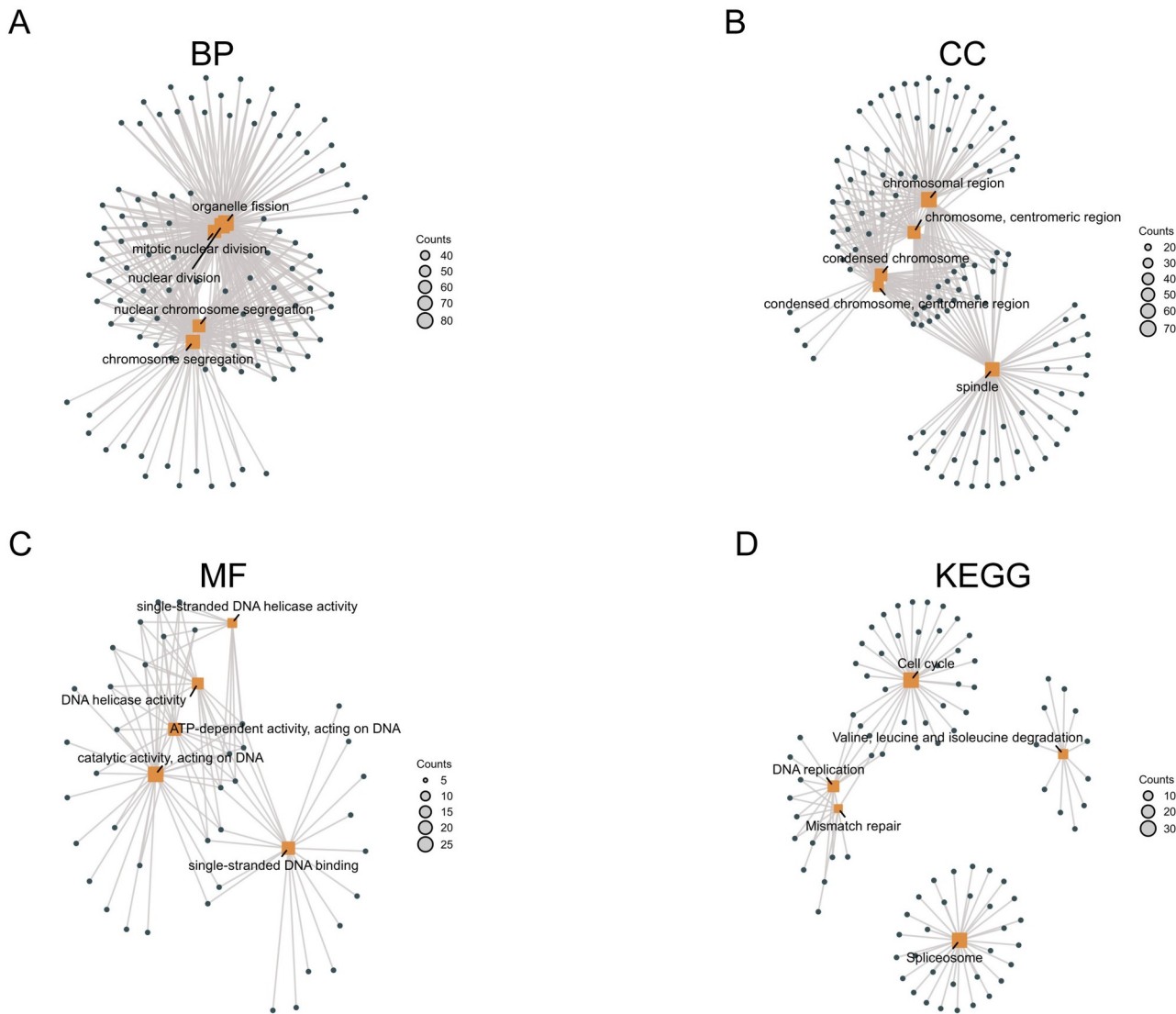

**Fig 11. GO and KEGG enrichment analysis of HMGs.** (A) Biological process (BP); (B) Molecular function (MF); (C) Cellular component; (D) Kyoto Encyclopedia of Genes and Genomes (KEGG).

The correlation between HMGA2 and genes such as IGF2BP2 is particularly noteworthy, as IGF2BP2 has been implicated in enhancing the stability of oncogenic mRNAs, thereby promoting tumorigenesis [37]. This suggests that HMGA2 may interact with IGF2BP2 to facilitate the expression of key oncogenes in HCC, contributing to tumor growth and progression. Similarly, CCNA2 is essential for cell cycle progression, and its association with HMGA2 may indicate a collaborative role in driving the proliferation of HCC cells [38].

The identification of these correlations not only provides insights into the molecular mechanisms by which HMGs may contribute to HCC but also highlights potential new avenues for therapeutic intervention. Targeting the interactions between HMGs and these correlated genes could disrupt critical pathways involved in HCC progression, offering novel strategies for treatment. Future studies should focus on functional validation of these interactions to confirm their roles in HCC and explore their potential as therapeutic targets.

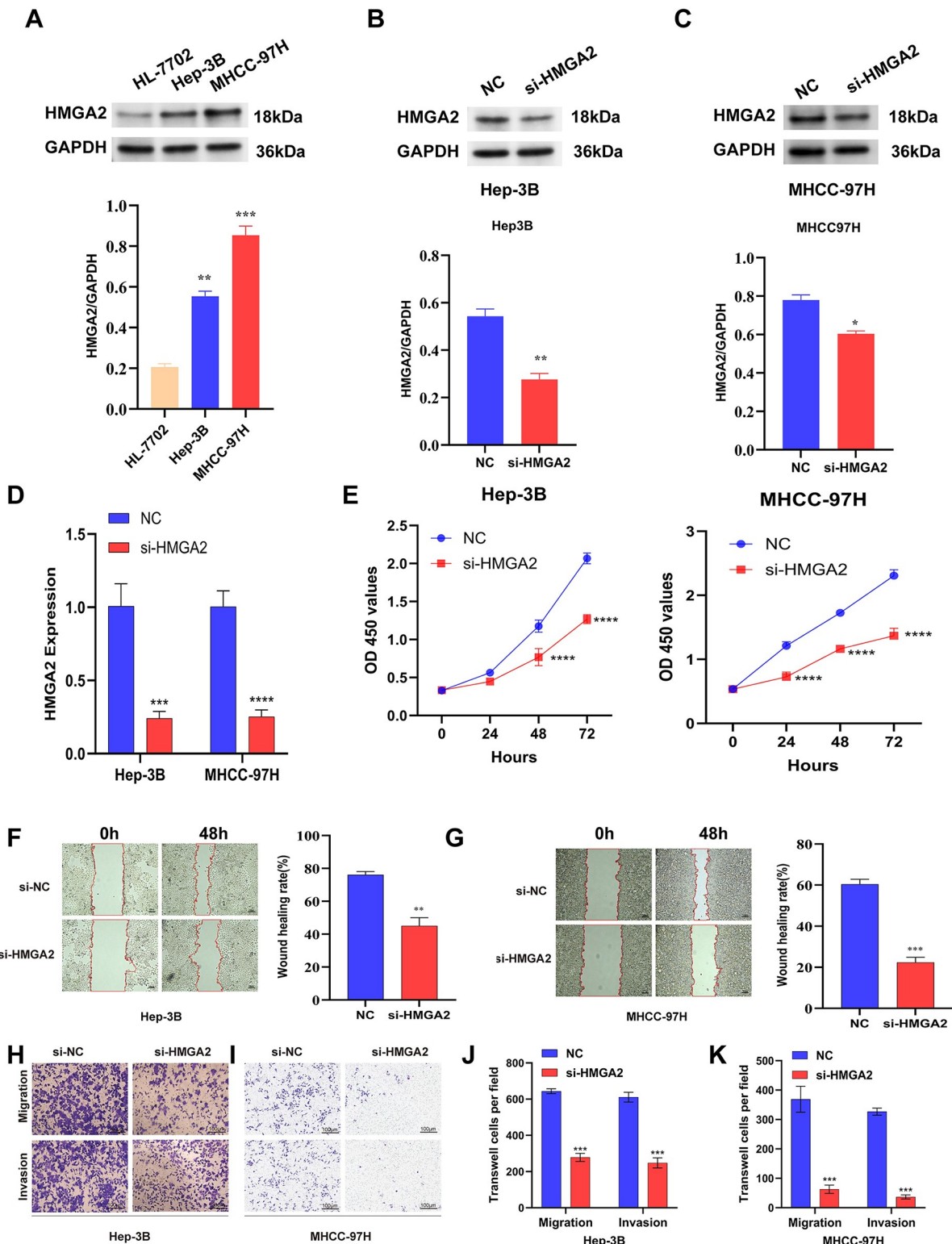

**Fig 12. HMGA2 expression is upregulated in tumour tissues and promotes tumour cell proliferation, migration and invasion.** (A) The protein expression levels of HMGA2 in human Normal liver cells and HCC cell lines, were detected by western blot analysis. (B-C) HMGA2 was silenced using siRNA technology. (D) Knockdown efficiency of HMGA2 in two Liver cancer cell lines Hep-3B and MHCC-97H, including the si-NC and si-HMGA2 groups. (E) CCK-8 assays were applied to detect the effect of HMGA2 knockdown on the proliferation of Hep-3B and MHCC-97H cell lines. (F,G) Wound-healing assays employed to detect the migration ability of HMGA2 knockdown cells, including MHCC-97H cell lines. (H-K) Transwell assays of Hep-3B and MHCC-97H cell lines. (*p < 0.05; **p < 0.01; ***p < 0.001).

By means of GO and KEGG enrichment analysis, we unveiled that HMGs prominently participate in vital cellular processes such as the cell cycle signaling pathway and DNA replication. Based on the above results and research, HMGA2 may play a crucial role in the development and progression of liver cancer cells. Further studies revealed that transfecting liver cancer cells with HMGA2 siRNA in vitro silenced the expression of HMGA2. The CCK-8 assay demonstrated that low levels of HMGA2 could inhibit the proliferation of liver cancer cells. Additionally, scratch and Transwell assays indicated that silencing HMGA2 in liver cancer cells significantly suppressed both horizontal and vertical migration abilities. These findings suggest that HMGA2 may play an important role in the growth and metastasis of liver cancer cells.

To date, numerous studies have shown that HMGA2 plays a crucial role in embryonic development, but when expressed in adult cells, it is regarded as an oncoprotein. HMGA2 enhances tumor cell invasiveness, metastasis, and chemoresistance by affecting cellular biological processes, leading to the failure of many cancer treatments. It is also one of the main reasons for the reduced survival rates of cancer patients. Increasing evidence indicates that HMGA2 is an independent prognostic marker for malignant tumors and that the expression level of HMGA2 is related to the efficacy of certain chemotherapeutic drugs. Existing research has confirmed the close association of HMGA2 with the occurrence, progression, metastasis, and prognosis of many malignant tumors [39, 40] this includes its pivotal role in tumor invasion and metastasis, as evidenced in malignancies such as breast cancer, prostate cancer, and colorectal cancer. Zhu et al [41] found that upregulation of HMGA2 promotes EMT in laryngeal squamous cell carcinoma. Huang et al [42] found that elevated levels of HMGA2 have been potentially linked to diminished overall survival among individuals diagnosed with head and neck cancer, hepatocellular carcinoma, renal cancer, and pancreatic ductal adenocarcinoma. The qRT-PCR findings unveiled notable upregulation of HMGA2 within HCC cells. Notably, heightened HMGA2 expression displays a significant correlation with diminished disease-specific survival (DSS), alongside its association with distinct clinical staging. With ongoing research, HMGA2 has been recognized as being extensively involved in the development and progression of various malignant tumors. Its expression levels are closely related to the clinicopathological characteristics of cancer patients, such as lymph node metastasis and TNM staging. To date, numerous studies have shown that HMGA2 expression is dysregulated in different human tumor tissues, and its high expression may be a critical step in the pathogenesis of malignant tumors. At the genetic level, knocking out the HMGA2 gene in cancer cells can induce apoptosis, cause cell cycle arrest, and inhibit tumor migration and invasion. At the protein level, HMGA2 can serve as a prognostic indicator for cancer patients.Therefore, detecting the expression levels of HMGA2 in tumor tissues can provide clinicians with important prognostic data, but further research is needed to understand how it affects tumor prognosis. In-depth studies on the crucial mechanisms of HMGA2 in tumor development and progression are of great significance for the development of HMGA2-targeted therapies.

In summary, this study utilized bioinformatics to investigate the expression patterns of HMGs in liver cancer, analyzed their possible mechanisms involved in liver cancer development, and conducted preliminary validation through experimental methods, providing a reference for further exploration of the oncogenic mechanisms of HMGs. The specific regulatory modes of each member of the HMG family and their involvement in signaling pathways in liver cancer cells are worth further exploration.

## Conclusion

Overall, this study reveals the oncogenic roles of most HMG family members in HCC, except for HMGN5, and highlights HMGA2 as a potential novel independent prognostic biomarker.

Our study offers a fresh vantage point for selecting prognostic biomarkers associated with HMGs in HCC, with its forthcoming application potentially guiding the formulation of optimal strategies for the management of our patients.

## Supporting information

**S1 Table. Univariate and multivariate analyses of clinicopathological variables and HMGs expressions for prediction of OS of TCGA patients.**
(DOCX)

**S2 Table. Univariate and multivariate analyses of clinicopathological variables and HMGs expressions for prediction of DSS of TCGA patients.**
(DOCX)

**S3 Table. Univariate and multivariate analyses of clinicopathological variables and HMGs expressions for prediction of PFI of TCGA patients.**
(DOCX)

**S1 Fig. ROC curves were plotted to validate the diagnostic efficacy of HMGs using GSE84402 and GSE76427.**
(DOCX)

**S1 File. Contains the original uncropped images underlying all blot results.**
(DOCX)

## Acknowledgments

We gratefully acknowledge the authors and participants of all datasets used in this study for their invaluable contributions. We also extend our thanks to Xiantaozi (www.xiantao.love) platform for providing the platform utilized in part of our data analysis.

## Author Contributions

**Conceptualization:** Xiangjie Liu.

**Data curation:** Xiangjie Liu.

**Formal analysis:** Qiangqiang Zhong, Baokang Zhao.

**Funding acquisition:** Xiangjie Liu.

**Investigation:** Qiangqiang Zhong, Baokang Zhao, Xiao She.

**Methodology:** Qiangqiang Zhong, Baokang Zhao, Xiao She.

**Project administration:** Qiangqiang Zhong, Baokang Zhao, Xiao She.

**Resources:** Xiangjie Liu.

**Software:** Qiangqiang Zhong, Baokang Zhao, Xiao She.

**Supervision:** Xiangjie Liu.

**Validation:** Qiangqiang Zhong, Baokang Zhao, Xiao She.

**Visualization:** Qiangqiang Zhong, Baokang Zhao.

**Writing – original draft:** Qiangqiang Zhong.

**Writing – review & editing:** Qiangqiang Zhong, Baokang Zhao, Xiao She, Xiangjie Liu.

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
