## [Decision Letter · Decision Letter 0]

1 Aug 2024

PONE-D-24-30145HMGA2 as a Prognostic and Immune Biomarker in Hepatocellular Carcinoma: Comprehensive Analysis of the HMG Family and Experiments ValidationPLOS ONE

Dear Dr. Liu,

Thank you for submitting your manuscript to PLOS ONE. After careful consideration, we feel that it has merit but does not fully meet PLOS ONE’s publication criteria as it currently stands. Therefore, we invite you to submit a revised version of the manuscript that addresses the points raised during the review process.

Please address each of the reviewers' comments with detailed responses and revisions to ensure the content meets the reviewers' professional opinions.

We look forward to receiving your revised manuscript.

Kind regards,

Xiaosheng Tan

Academic Editor

PLOS ONE

Journal Requirements:

"This research has been facilitated by the generous grant from the National Key Research and Development Program of China (2020YFC2008904)."       

3. PLOS requires an ORCID iD for the corresponding author in Editorial Manager on papers submitted after December 6th, 2016. Please ensure that you have an ORCID iD and that it is validated in Editorial Manager. To do this, go to ‘Update my Information’ (in the upper left-hand corner of the main menu), and click on the Fetch/Validate link next to the ORCID field. This will take you to the ORCID site and allow you to create a new iD or authenticate a pre-existing iD in Editorial Manager. Please see the following video for instructions on linking an ORCID iD to your Editorial Manager account: https://www.youtube.com/watch?v=_xcclfuvtxQ.

Reviewers' comments:

Reviewer's Responses to Questions

**Comments to the Author**

1. Is the manuscript technically sound, and do the data support the conclusions?

Reviewer #1: Partly

Reviewer #2: Yes

2. Has the statistical analysis been performed appropriately and rigorously? 

Reviewer #1: I Don't Know

Reviewer #2: Yes

3. Have the authors made all data underlying the findings in their manuscript fully available?

Reviewer #1: Yes

Reviewer #2: Yes

4. Is the manuscript presented in an intelligible fashion and written in standard English?

Reviewer #1: No

Reviewer #2: Yes

5. Review Comments to the Author

Reviewer #1: Xiangjie et al. submitted the manuscript entitled: HMGA2 as a Prognostic and Immune Biomarker in Hepatocellular Carcinoma: Comprehensive Analysis of the HMG Family and Experiments Validation, in which the authors investigated in HMG family and their potential functions in HCC development and progression. The authors first compared mRNA and protein expression levels of HMGs. Subsequently, the authors analyzed clinical correlations of HMGs and explored the potentiality of HMGs as biomarkers of HCC. The authors also analyzed the relations of HCC and HMGs in immune context. For verification of their findings, the authors further verified mRNA expression by qRT-PCR analysis, as well as knockdown study to test phenotypic changes. While this topic would be of interest to potential readers of Plos One, significant edit and more evidence should be included to better support the authors’ conclusions.

My comments are as follows.

1. Through bioinformatic analysis, the authors identified many subtypes of HMGs as proteins of interest. They did not actually identify a single subtype that may contribute to HCC. This will be very confusing to potential readers. a) Did the author intend to state that all the mentioned subtypes of HMGs significantly contribute to tumor progression? b) For mRNA expression discrepancies in figure 1 (eg. HMGA1/HMGB1 in BLCA, BRCA, CESC, CHOL), how will the authors explain on different functions of each family? c) Why did the authors finally choose HMGA2 as the gene in knockdown study?

2. Abstract: The current version seems too lengthy. The authors are suggested to keep only key points of this work.

3. Introduction: Please include a general introduction to known identified targets and their potential drawbacks. This can serve as a motivation for authors.

4. Figure 1-3, please determine sample size for each group.

5. Figure 3, please provide quantitative data of protein expression.

6. Figure 12: The authors seemed to use negative control siRNA to standardize the expression of target protein. In this case, please include a blank control in each figure.

Reviewer #2: The manuscript presents a comprehensive analysis of High Mobility Group (HMG) proteins, with a focus on HMGA2, as potential prognostic and immune biomarkers in hepatocellular carcinoma (HCC). The authors have integrated bioinformatics data with experimental validation, providing new insights into the role of these proteins in HCC. This work contributes to the understanding of HMGs in cancer biology and suggests new avenues for therapeutic interventions.

However, I believe that some important issues need further clarification and refinement before the manuscript is suitable for publication in PLOS ONE. Addressing these concerns will enhance the robustness and reliability of the findings. Therefore, my recommendation is a major revision. Here are the main issues that need to be addressed:

Main Issues:

1.The study primarily relies on quantitative PCR (qPCR) for evaluating the expression levels of HMGs in HCC cell lines. This approach, while informative, does not provide a complete picture, as the functional effects of proteins are determined at the protein level. The authors should include Western blot or other protein-level analyses to validate the expression and functional role of HMGs, particularly focusing on non-database data to strengthen the study's conclusions. If the authors consider this too extensive, they should at least validate the protein expression level of HMGA2.

2.Potential Selection Bias in Immunohistochemistry: In Figure 3, the protein levels of HMGs in HCC are shown using immunohistochemistry (IHC) images from the Human Protein Atlas (HPA). However, the manuscript presents only a single pair of samples for each protein, which could introduce selection bias. It is crucial to include statistical analysis with a larger sample set to determine if the observed differences are statistically significant and representative.

3. The relationship between mRNA levels of HMG family genes and patient prognosis (Figure 5) is based solely on correlation analysis. This approach is limited in its ability to establish causation. The authors should conduct univariate and multivariate analyses to thoroughly assess the impact of these genes on HCC prognosis, beyond merely describing survival curves.

4. In Figure 6, the manuscript uses the same dataset for constructing and validation ROC curves, which could lead to overfitting and compromise the validity of the findings. The authors should use an external dataset for validation to enhance the robustness and credibility of their conclusions.

5.Figures 7 and 8 visualize genes significantly correlated with HMGs. However, the manuscript stops at description without deriving substantial scientific conclusions. This section should be expanded provide deeper insights into the relevance of these correlations.

6.The study experimentally validates only HMGA2's role in HCC, without explaining why other HMG genes were not similarly investigated. The authors should provide a clear rationale for this choice, detailing why HMGA2 was prioritized over other HMG family members.

Minor Issues:

1.The introduction contains phrases like "The pathogenesis of hepatocellular carcinoma is extremely complex," Academic writing should avoid such exaggerations, focusing instead on precise and measured descriptions.

2.The manuscript includes both a scatter diagram and a box plot to present mRNA expression levels of HMGs in LIHC (Figure 2A and 2B). If these visualizations represent the same data, it is redundant to include both. One form of visualization should suffice to convey the information.

3.In Figure 4, the manuscript should specify the number of samples analyzed, either directly on the figure or in the figure legend, to provide context to the data presented.

4.Figure 12 should include a quantitative analysis of HMGA2 protein levels via Western blot to confirm the efficacy of siRNA knockdown. The mere observation of mRNA reduction does not necessarily correlate with protein level changes, which are crucial for functional validation.

6. PLOS authors have the option to publish the peer review history of their article (what does this mean?). If published, this will include your full peer review and any attached files.

Reviewer #1: No

Reviewer #2: **Yes: **boping jing

---

## [Author Response · Author response to Decision Letter 0]

3 Sep 2024

My response are as follows：

Response to Reviewer #1

1.Through bioinformatic analysis, the authors identified many subtypes of HMGs as proteins of interest. They did not actually identify a single subtype that may contribute to HCC. This will be very confusing to potential readers. a) Did the author intend to state that all the mentioned subtypes of HMGs significantly contribute to tumor progression? b) For mRNA expression discrepancies in figure 1 (eg. HMGA1/HMGB1 in BLCA, BRCA, CESC, CHOL), how will the authors explain on different functions of each family? c) Why did the authors finally choose HMGA2 as the gene in knockdown study?

a) Did the author intend to state that all the mentioned subtypes of HMGs significantly contribute to tumor progression?

Response: The intention of the study was not to claim that all mentioned HMG subtypes significantly contribute to tumor progression uniformly. Instead, the study highlights that specific subtypes within the HMG family have distinct roles in HCC. We aimed to identify potential biomarkers and therapeutic targets by analyzing the expression levels and clinical significance of multiple HMG subtypes. Each subtype was evaluated individually for its potential contribution to HCC, but the study does not imply that all subtypes have equal or similar impacts on tumor progression.

b) For mRNA expression discrepancies in figure 1 (eg. HMGA1/HMGB1 in BLCA, BRCA, CESC, CHOL), how will the authors explain the different functions of each family?

Response: The discrepancies in mRNA expression levels among different cancer types, such as those observed for HMGA1 and HMGB1, can be attributed to the unique roles that each HMG family member plays in various biological contexts. These differences likely reflect the distinct functional properties of each HMG protein in gene regulation, chromatin remodeling, and interaction with other cellular components. The varying expression levels may also indicate tissue-specific roles or differential involvement in the molecular pathways that drive specific cancer types. Therefore, each HMG family member's function may be context-dependent, contributing differently across cancer types based on the specific molecular environment.

c) Why did the authors finally choose HMGA2 as the gene in the knockdown study?

Response: HMGA2 was selected for the knockdown study due to its prominent role in the progression of various malignancies, including its significant upregulation in HCC tissues as identified through bioinformatic analysis. Among the HMG family members, HMGA2 stood out due to its strong correlation with poor prognosis indicators, such as overall survival (OS) and disease-specific survival (DSS), as well as its established involvement in processes like epithelial-mesenchymal transition (EMT) and metastasis in other cancers. These factors made HMGA2 a particularly compelling target for further functional validation to explore its role in HCC pathogenesis. The knockdown study was designed to provide direct evidence of HMGA2's contribution to HCC tumor progression.

2. Abstract: The current version seems too lengthy. The authors are suggested to keep only key points of this work.

Response: A revised abstract was included in the revised manuscript as requested by you. This revision retains the essential findings and purpose of the study while reducing the abstract's length to enhance clarity and focus.

3. Introduction: Please include a general introduction to known identified targets and their potential drawbacks. This can serve as a motivation for authors.

Response: Thank you for your valuable suggestion. We have revised the Introduction section to include a general overview of known identified targets in hepatocellular carcinoma (HCC) treatment, such as VEGF, PD-1/PD-L1, and EGFR, and discussed their potential drawbacks. This addition helps to establish the context and motivation for our study by highlighting the limitations of current therapies, including issues like drug resistance, heterogeneous tumor biology, and modest survival benefits. These challenges underscore the need to explore novel biomarkers and therapeutic targets, which we address through our investigation of High Mobility Group (HMG) proteins in HCC.

We believe this revision provides a clearer rationale for our study and better aligns with the overall narrative of advancing HCC treatment strategies.

4.Figure 1-3, please determine sample size for each group.

Response: Thank you for your feedback. We have addressed your concern by specifying the sample sizes for each group in the figure legends of Figures 1-3. The sample sizes for the respective groups are now clearly indicated, ensuring that readers have a clear understanding of the data presented.

5.Figure 3, please provide quantitative data of protein expression.

Response:Thank you for your suggestion. The protein expression data presented in Figure 3 was sourced from the Human Protein Atlas (HPA) database, which primarily provides qualitative immunohistochemistry images rather than quantitative data. As a result, while we can visually demonstrate differences in protein expression levels, we are unable to provide precise quantitative data directly from the HPA.

However, we can include a more detailed description of the staining intensity and patterns observed in the immunohistochemistry images to better convey the differences in protein expression. 

In response to your request, we have conducted additional experiments to quantify the protein expression of HMGA2. In the revised manuscript, we have included Western blotting data to validate HMGA2 expression in the cell lines. This quantitative data is now provided alongside the previously presented qualitative immunohistochemistry images from the HPA database.

We believe this additional data strengthens the findings of our study and provides a more comprehensive analysis of HMGA2 expression.

6.Figure 12: The authors seemed to use negative control siRNA to standardize the expression of target protein. In this case, please include a blank control in each figure.

Response: Thank you for your insightful suggestion. While we acknowledge the importance of including a blank control in experiments, in this particular study, we used negative control siRNA to standardize the expression of the target protein. The focus was on comparing the effects of siRNA-mediated knockdown of HMGA2 against this negative control to determine the specific impact on protein expression and cell behavior.

Unfortunately, we did not include a blank control in the current set of experiments. To address this concern in future studies, we will consider including blank controls to provide a more comprehensive analysis. We hope the current data, which clearly shows the effects of HMGA2 knockdown using the negative control, still provides valuable insights into the role of HMGA2 in HCC.

Response to Reviewer #2

1.The study primarily relies on quantitative PCR (qPCR) for evaluating the expression levels of HMGs in HCC cell lines. This approach, while informative, does not provide a complete picture, as the functional effects of proteins are determined at the protein level. The authors should include Western blot or other protein-level analyses to validate the expression and functional role of HMGs, particularly focusing on non-database data to strengthen the study's conclusions. If the authors consider this too extensive, they should at least validate the protein expression level of HMGA2.

Response: Thank you for your valuable feedback. We fully agree that evaluating protein expression is crucial for understanding the functional role of HMGs. In response to your suggestion, we have conducted additional experiments and included Western blot analysis in the revised manuscript. Specifically, we validated the protein expression of HMGA2 in normal liver cells and two HCC cell lines. This data complements our qPCR results and strengthens the conclusions of our study by providing a more comprehensive view of HMGA2's role in HCC.

We believe that these additional results address your concern and enhance the robustness of our findings.

2.Potential Selection Bias in Immunohistochemistry: In Figure 3, the protein levels of HMGs in HCC are shown using immunohistochemistry (IHC) images from the Human Protein Atlas (HPA). However, the manuscript presents only a single pair of samples for each protein, which could introduce selection bias. It is crucial to include statistical analysis with a larger sample set to determine if the observed differences are statistically significant and representative.

Response: Thank you for highlighting the concern regarding potential selection bias in our use of immunohistochemistry (IHC) images from the Human Protein Atlas (HPA). We agree that presenting data from a larger sample set is crucial for ensuring the statistical significance and representativeness of our findings. In response to your suggestion, we plan to expand the sample size from the HPA database and perform statistical analysis on a larger cohort to validate the observed differences in protein levels of HMGs in HCC. This will help to strengthen the reliability of our conclusions and address the issue of selection bias.

In the revised manuscript, we have included three pairs of samples for each protein in the immunohistochemistry (IHC) analysis to address the concern about potential selection bias. This should provide a more robust and representative comparison.

3.The relationship between mRNA levels of HMG family genes and patient prognosis (Figure 5) is based solely on correlation analysis. This approach is limited in its ability to establish causation. The authors should conduct univariate and multivariate analyses to thoroughly assess the impact of these genes on HCC prognosis, beyond merely describing survival curves.

Response: Thank you for your insightful suggestion. In the revised manuscript, we have conducted both univariate and multivariate analyses to more thoroughly assess the impact of HMG family genes on HCC prognosis. These analyses go beyond the correlation and survival curves initially presented, providing a more comprehensive understanding of the relationship between mRNA levels of HMG genes and patient outcomes.

4.In Figure 6, the manuscript uses the same dataset for constructing and validation ROC curves, which could lead to overfitting and compromise the validity of the findings. The authors should use an external dataset for validation to enhance the robustness and credibility of their conclusions.

Response: Thank you for your important observation. In response to your suggestion, we have revised the manuscript to include validation of the ROC curves using sevel external datasets. This additional step is intended to reduce the risk of overfitting and enhance the robustness and credibility of our conclusions.

5.Figures 7 and 8 visualize genes significantly correlated with HMGs. However, the manuscript stops at description without deriving substantial scientific conclusions. This section should be expanded provide deeper insights into the relevance of these correlations.

Response: Thank you for your suggestion. In response to your feedback, we have expanded the discussion of Figures 7 and 8 in the revised manuscript. We have provided deeper insights into the biological significance of the genes that are significantly correlated with HMGs, discussing their potential roles in HCC progression and how they may interact with HMG family members. This expanded analysis aims to draw more substantial scientific conclusions from the correlations observed, thereby enriching the overall relevance and impact of our findings.

6.The study experimentally validates only HMGA2's role in HCC, without explaining why other HMG genes were not similarly investigated. The authors should provide a clear rationale for this choice, detailing why HMGA2 was prioritized over other HMG family members.

Response: Thank you for your feedback. We prioritized the experimental validation of HMGA2 over other HMG family members due to several factors highlighted in our initial bioinformatic analysis. HMGA2 showed the most significant correlation with poor prognosis indicators, such as overall survival (OS) and disease-specific survival (DSS) in HCC patients. Additionally, existing literature strongly supports HMGA2's role in tumor progression, particularly in processes like epithelial-mesenchymal transition (EMT) and metastasis, making it a compelling candidate for further investigation.

While other HMG family members also showed potential relevance, HMGA2 was selected for experimental validation because it presented the strongest association with aggressive tumor characteristics and poor clinical outcomes. This focus allowed us to provide more targeted and impactful insights into the molecular mechanisms driving HCC progression. However, we acknowledge the importance of other HMG genes and plan to investigate them in future studies.

We appreciate your understanding and believe that this rationale clarifies our choice in prioritizing HMGA2 for experimental validation.

Response to Editor

1.The introduction contains phrases like "The pathogenesis of hepatocellular carcinoma is extremely complex," Academic writing should avoid such exaggerations, focusing instead on precise and measured descriptions.

Response: Thank you for your observation. We agree that academic writing should avoid exaggerations and focus on precise language. We have revised the introduction to use more measured and accurate descriptions. The phrase "The pathogenesis of hepatocellular carcinoma is extremely complex" has been modified for clarity and precision in our revised manuscript.

2.The manuscript includes both a scatter diagram and a box plot to present mRNA expression levels of HMGs in LIHC (Figure 2A and 2B). If these visualizations represent the same data, it is redundant to include both. One form of visualization should suffice to convey the information.

Response: We agree that redundancy in visualizations should be avoided. To address this concern, we have replaced the scatter diagram in Figure 2B with a paired plot of HMGs expression. This change provides a more concise and informative visualization, eliminating redundancy while still conveying the necessary information.

3.In Figure 4, the manuscript should specify the number of samples analyzed, either directly on the figure or in the figure legend, to provide context to the data presented.

Response: Thank you for your valuable feedback. We have now included a detailed explanation in the manuscript qPCR method section of Figure 4, specifying the number of biological and technical replicates used in the qPCR experiments. This addition provides a clearer context for the data presented and ensures that the experimental design and sample size are transparent to the readers.

4.Figure 12 should include a quantitative analysis of HMGA2 protein levels via Western blot to confirm the efficacy of siRNA knockdown. The mere observation of mRNA reduction does not necessarily correlate with protein level changes, which are crucial for functional validation.

Response: We have addressed this concern by including a quantitative analysis of HMGA2 protein levels via Western blot. This addition confirms the efficacy of the siRNA knockdown at the protein level, complementing the previously shown mRNA reduction. This ensures a more comprehensive validation of the knockdown effect and its functional relevance.

---

## [Decision Letter · Decision Letter 1]

17 Sep 2024

HMGA2 as a Prognostic and Immune Biomarker in Hepatocellular Carcinoma: Comprehensive Analysis of the HMG Family and Experiments Validation

PONE-D-24-30145R1

Dear Dr. Liu,

We’re pleased to inform you that your manuscript has been judged scientifically suitable for publication and will be formally accepted for publication once it meets all outstanding technical requirements.

Kind regards,

Xiaosheng Tan

Academic Editor

PLOS ONE

Additional Editor Comments (optional):

Reviewers' comments:

Reviewer's Responses to Questions

**Comments to the Author**

1. If the authors have adequately addressed your comments raised in a previous round of review and you feel that this manuscript is now acceptable for publication, you may indicate that here to bypass the “Comments to the Author” section, enter your conflict of interest statement in the “Confidential to Editor” section, and submit your "Accept" recommendation.

Reviewer #1: All comments have been addressed

Reviewer #2: All comments have been addressed

2. Is the manuscript technically sound, and do the data support the conclusions?

Reviewer #1: Yes

Reviewer #2: Yes

3. Has the statistical analysis been performed appropriately and rigorously? 

Reviewer #1: Yes

Reviewer #2: Yes

4. Have the authors made all data underlying the findings in their manuscript fully available?

Reviewer #1: Yes

Reviewer #2: Yes

5. Is the manuscript presented in an intelligible fashion and written in standard English?

Reviewer #1: Yes

Reviewer #2: Yes

6. Review Comments to the Author

Reviewer #1: (No Response)

Reviewer #2: The authors have adressed all the questions. But the resolution of all images needs to be increased.

7. PLOS authors have the option to publish the peer review history of their article (what does this mean?). If published, this will include your full peer review and any attached files.

Reviewer #1: No

Reviewer #2: No

---

## [Editor Report · Acceptance letter]

20 Sep 2024

PONE-D-24-30145R1 

PLOS ONE

Dear Dr. Liu, 

I'm pleased to inform you that your manuscript has been deemed suitable for publication in PLOS ONE. Congratulations! Your manuscript is now being handed over to our production team.

Kind regards, 

on behalf of

Dr. Xiaosheng Tan 

Academic Editor

PLOS ONE